# Fire, environmental and anthropogenic controls on pantropical tree cover
Douglas I. Kelley [1] ✉, France Gerard[1], Ning Dong [2,3] ✉, Chantelle Burton [4], Arthur Argles[4], Guangqi Li[5,6], Rhys Whitley[7], Toby R. Marthews[1], Eddy Roberston[4], Graham P. Weedon[8], Gitta Lasslop[9], Richard J. Ellis[1], Ioannis Bistinas[10] & Elmar Veenendaal [11]

Explaining tropical tree cover distribution in areas of intermediate rainfall is challenging, with fire's role in limiting tree cover particularly controversial. We use a novel Bayesian approach to provide observational constraints on the strength of the influence of humans, fire, rainfall seasonality, heat stress, and wind throw on tropical tree cover. Rainfall has the largest relative impact on tree cover (11.6–39.6%), followed by direct human pressures (29.8–36.8%), heat stress (10.5–23.3%) and rainfall seasonality (6.3–22.8%). Fire has a smaller impact (0.2–3.2%) than other stresses, increasing to 0.3–5.2% when excluding human influence. However, we found a potential vulnerability of eastern Amazon and Indonesian forests to fire, with up to 2% forest loss for a 1% increase in burnt area. Our results suggest that vegetation models should focus on fire development for emerging fire regimes in tropical forests and revisit the linkages between rainfall, non-fire disturbances, land use and broad-scale vegetation distributions.

While precipitation gradient explains much of the transition between tropical forests and more open ecosystems[1,2], many different vegetation and tree compositions exist in intermediate and particularly seasonal precipitation regions. Beyond agricultural and urban areas, which have replaced or degraded ~29.2% of natural tropical ecosystems (Supplementary Fig. 1), many vegetation models invoke fire as a primary disturbance to reproduce realistic vegetation distribution in seasonal tropical climates, maintaining low tree cover in savanna and grassland ecosystems[3–7]. In these Fire Enabled Dynamic Global Vegetation Models (FDGVMs), fire maintains a balance between woody and non-woody vegetation by preventing the encroachment of trees into grasslands and promoting the growth of more fire-adapted herbaceous plant types. FDGVMs consider fire a significant disturbance in most tropical ecosystems[8]. While extreme temperatures can reduce tree cover by causing water stress, leading to leaf area reduction, branch cavitation and ultimately, death of trees, models generally simulate a larger impact from fire than heat stress on tropical tree cover, particularly in savannas and grasslands[9].

However, there is a lack of empirical or observational data or studies that can directly inform the relative importance of these controls on FDGVM resolutions across the tropics[10] and modelled fire impacts on vegetation are particularly poorly constrained observationally[7,11,12]. Field-based[13] and empirical analyses using remotely sensed products identify the coincidence of burnt area with the reduced occurrence ("missing") of intermediate (50–60%) tree covers[14–18] as evidence of a substantial impact of fire on tree cover. On the other hand, recent field analyses of fire impacts in relation to soil and climate[19,20] suggest that fire could have a smaller effect on tropics-wide tree cover than suggested up to now. Critical evaluations of the most used global woody cover product have also questioned the intermediate tree cover gap[21–24]. Finding constraints on the relative contributions of different factors in maintaining tree cover has become particularly important given how much global vegetation models are utilised to assess ecosystem and carbon impacts of climate, carbon budgets, and environmental and land use change[25,26].

For this analysis, we focus on the "maintenance of tree cover"—the factors that support or suppress existing tree cover. This approach examines what sustains the current tree cover rather than the dynamic changes in tree cover over time. It differs from the "determination" of tree cover, which

[1]UK Centre for Ecology and Hydrology, Wallingford, OX10 8BB, UK. [2]College of Resources and Environment, Huazhong Agricultural University, Wuhan, 430070, China. [3]Georgina Mace Centre for the Living Planet, Imperial College London, Department of Life Sciences, Silwood Park Campus, Ascot, SL5 7PY, UK. [4]Met Office Hadley Centre for Climate Science and Services, Exeter, UK. [5]Department of Geography and Environmental Science, University of Reading, Reading, UK. [6]Biological and Environmental Sciences, University of Stirling, Stirling, UK. [7]Natural Perils Pricing, Consumer Insurance, Suncorp, Sydney, Australia. [8]Met Office, Joint Centre for Hydro-Meteorological Research (JCHMR), Crowmarsh Gifford, Wallingford, Oxfordshire, OX10 8BB, UK. [9]Senckenberg Biodiversity And Climate Research Centre, Frankfurt, Germany. [10]Cognizant Benelux BV, Paul van Vlissingenstraat 10, 1096BK Amsterdam, The Netherlands. [11]Plant Ecology and Nature Conservation Group, Wageningen University, Wageningen, The Netherlands. ✉e-mail: doukel@ceh.ac.uk; n.dong@mail.hzau.edu.cn

reflects both historical and present conditions, such as how fires might clear land and transition forests to different land uses[27,28].

We employed a Bayesian limitation framework[29,30] to test climate and environmental influences on tropical tree cover. The framework was optimised against tree cover observations from the MODIS Vegetation Continuous Fields (VCF) collection 6[31] at a spatial resolution of 0.5° (~50 km at the equator), similar to standard FDGVM resolutions[32,33], noting that the main Fire Modelling Intercomparison Project (FireMIP) is moving to a 0.5° grid as standard[34]. The framework models tree cover as a product of four limiting factors (see Supplementary Fig. 1; Supplementary Table 2):

1. Mean Annual Precipitation (MAP)
2. Energy, combining Mean Annual Temperature (MAT) and Shortwave Radiation (SW)
3. Stress or disturbance, which includes factors such as fire, rainfall seasonality, heat, and windthrow
4. Human pressure, expressed as land use (cropland, pasture, urban cover) and population density.

Each limiting factor was modelled as a linear combination of its drivers and represented by a logistic curve that takes a value between 0 and 1. The final Fractional Tree Cover (TC) was calculated as the product of limitations imposed by all four factors. This means, for instance, that MAP defines the upper limit on tree cover in regions where energy, stress, and human pressure are not limiting.

We applied a Bayesian inference technique to optimise the framework against MODIS VCF tree cover data[31]. The modelling approach allows us to systematically remove the influence of each control, enabling the exploration of how each factor independently affects tree cover. The Bayesian optimisation generates probability distributions for each simulated tree cover scenario, providing comprehensive uncertainty quantification[29,30]. This combination of systematically isolating the influence of each control and the Bayesian technique's production of probability distributions allows us to assess how confident we are in the model's predictions[30,35,36]. It also helps gauge the reliability of any experiments conducted, particularly when examining correlated factors like tropical fires and seasonal drought-induced stress[37,38], and co-varying land use and fire[7,39,40]. The Bayesian framework also inherently accounts for stochasticity, enabling the exploration of unpredictable factors, such as windthrow and long-return interval fires, and their potential impacts on tree cover[41]. This process allows for experiments that test the effects of removing or interacting drivers, making it possible to assess both individual and joint impacts of different factors on tree cover.

The optimisation was conducted across the pan-tropics between 30° North to South, leveraging the spatial variability of tree cover in MODIS VCF data to inform the model. To evaluate the model's performance, we used 20% of the MODIS VCF data for parameter optimisation and reserved the remaining 80% as validation data. While this overlap between training and validation datasets introduces some potential bias, the lack of alternative global datasets at this resolution makes this approach necessary. Importantly, we focused on understanding the relative impacts of different controls on tree cover, rather than producing absolute predictions. This also mitigates against any potential biases in the MODIS VCF training data[21–24]. MODIS VCF data was used to calibrate the model but was not incorporated as input data in tree cover predictions. The Bayesian optimization generated parameter distributions that inform the final predictions, ensuring that the impacts of climate, stress, and human pressures on tree cover can be quantified independently of the direct use of MODIS VCF data in the final predictions.

The simulated distribution of tree cover compares well against test observations (see evaluation supplement and Supplementary Fig. 2, Supplementary Fig. 3). Where observations of tree cover align with the predicted range (Supplementary Fig. 4), indicates the model correctly reproduces the controls' influence on tree cover and its uncertainty[30]. We report changes in cover at the 10–90% percentile confidence range of the framework's posterior probability distribution, which provides a range of plausible

constraints on each factor's impact. Assumptions about noninteracting factors are included in uncertainty ranges[41]. See methods for full framework description.

When analysing the impact of different factors on tree cover, we consider two key measurements. The first is the absolute difference (or impact) on tree cover with and without the influence of the factor in question. The second is the relative impact on tree cover, measured as the difference in tree cover as a percentage of the original tree cover before the factor was introduced.

After assessing how large-scale climate gradients influence tree cover, we use this framework to test how stresses and human impacts limit tree cover tropic-wide and in different vegetation types. We identify fire impact on tree cover as a significantly lower response than found in fire-enabled DGVMs. We demonstrate how our results are consistent with field measurements and fire exclusion experiments before discussing the implications for global vegetation modelling. Finally, we explore where tree cover is sensitive to recent or potential future changes in fire.

## Results
### Environmental controls and human impact
MAP is the primary control of tropical tree cover and dominates in arid and semi-arid ecosystems, with rainfall distribution consistently the largest factor influencing stress (Table 1, Fig. 1, Supplementary Fig. 5). Limitation from MAP reduces relative tree cover by 11.6–39.6% (Table 1). MAP exerts the least control in wet forests (2.4–13.1%) and most in deserts (35.3–72.0%). Energy (i.e., MAT and SW combined) only has a small impact on tropical tree cover, primarily from co-limitation with MAP in mountainous areas (Fig. 1), impacting relative tree cover by 0–0.09% (Table 1).

Limitation from environmental stresses (i.e., the combined stresses of rainfall seasonality, fire, heat stress and wind) occurs almost everywhere in the tropics, resulting in a 12.8–28.6% relative tree cover reduction (Table 1). We use four metrics as a proxy for rainfall seasonality: Fractional mean annual dry days, fractional number of dry days in the driest month, precipitation in the driest month, and mean seasonal precipitation concentration. The results presented here summarize the impact via performance-weighted contributions of all. Rainfall seasonality has the most considerable impact from all the stress controls, causing a relative tree cover reduction of 10.5–23.3%. It is particularly important in the seasonal, fire-prone savanna ecosystems (reducing relative tree cover by 19.5–30.3%) in Africa, Asia, Indonesia, Australia and Southern Amazonia, in the savanna-forest ecotone areas extending into the Southern Amazonian forest and the tropical forests of Central Americas (Fig. 2). These stresses co-limit with MAP over much of the tropics but dominate in some African and Australian savannas (Fig. 1).

Limitation from human pressure (Land use and population) is important in southern Amazonia forests, Northern Andes and Chocó-Manabí Corridor and central tropical Indonesia, impacting relative tree cover by 29.8–36.8% (Table 1). All controls except energy limits tree cover in deciduous ecosystems. They are particularly affected by heat stress and rainfall seasonality, impacting relative tree cover by 12.1–34.6% and 21.0–35.4% respectively (Table 1).

### The impact of fire relative to other stresses
Overall, fire (assessed using burnt area) has the least impact from the stressors tested, with a relative tree cover reduction of 0.20–3.2%. Its impact is smaller than heat stress (relative reduction of 6.31–22.8%, Table 1, $p = 0.0001$, Supplementary Table 1) and windthrow (3.0–8.7%, $p = 0.061$). Between ecosystems, fire impact is greatest in savanna/grassland (0.6–7.1%), though this is still less than rainfall seasonality, heat stress and windthrow (Fig. 3; $P = 0.013, 0.020, 0.28$, respectively). Most of the fire impact is in African savannas (Fig. 2).

The fire's impact is concentrated in warm, seasonal climates with moderate rainfall, predominantly the same regions of high burnt area (Fig. 4). Where MAP is between 100–2000 mm/yr, fire impact on tree cover increases as rainfall seasonality and MTWM increase. Fire only impacts tree

**Table 1 | Environmental stresses and anthropogenic pressures impact on tropical tree cover by biome**

| | | All tropics | | Deciduous vegetation | | Wet Forests | | Dry Forest | | Savanna/ grass | | Mediterranean | | Summergreen Forests/ woodland | | Desert/shrub | |
|---|---|---|---|---|---|---|---|---|---|---|---|---|---|---|---|---|---|
| | | *10%* | *90%* | *10%* | *90%* | *10%* | *90%* | *10%* | *90%* | *10%* | *90%* | *10%* | *90%* | *10%* | *90%* | *10%* | *90%* |
| **Observations** | | 27.84 | | 17.54 | | 53.85 | | 35.76 | | 22.64 | | 38.7 | | 38.27 | | 13.43 | |
| **Simulation** | | 23.21 | 32.48 | 16.23 | 26.1 | 42.27 | 53.93 | 22.72 | 35.87 | 18.79 | 29.95 | 22.28 | 35.05 | 22.64 | 35.82 | 8.18 | 14.63 |
| **MAP** | area | 3.04 | 21.29 | 3.77 | 22.4 | 1.02 | 8.15 | 1.8 | 13.91 | 2.9 | 16.01 | 1.75 | 15.85 | 1.93 | 16.41 | 4.47 | 37.54 |
| | relative | 11.58 | 39.59 | 18.85 | 46.19 | 2.36 | 13.12 | 7.34 | 27.95 | 13.37 | 34.84 | 7.29 | 31.14 | 7.86 | 31.42 | 35.33 | 71.96 |
| **Energy** | area | 0 | 0.03 | 0 | 0.02 | 0 | 0.05 | 0 | 0.03 | 0 | 0.02 | 0 | 0.03 | 0 | 0.03 | 0 | 0.01 |
| | relative | 0 | 0.09 | 0 | 0.07 | 0 | 0.1 | 0 | 0.08 | 0 | 0.07 | 0 | 0.09 | 0 | 0.09 | 0 | 0.07 |
| **Env. Stress** | area | 3.4 | 12.98 | 5.47 | 17.01 | 2.22 | 12.07 | 2.61 | 11.57 | 5.47 | 17.15 | 0.78 | 6.98 | 0.9 | 8.44 | 0.66 | 5.07 |
| | relative | 12.77 | 28.55 | 25.2 | 39.46 | 4.99 | 18.29 | 10.31 | 24.39 | 22.56 | 36.41 | 3.38 | 16.61 | 3.81 | 19.08 | 7.44 | 25.75 |
| **Human Pressure** | area | 9.87 | 18.91 | 9.39 | 19.36 | 14.71 | 25.85 | 10.15 | 21.87 | 9.83 | 20.41 | 8.07 | 17.79 | 8.66 | 19.02 | 3.43 | 8.31 |
| | relative | 29.84 | 36.79 | 36.65 | 42.59 | 25.81 | 32.4 | 30.87 | 37.88 | 34.35 | 40.52 | 26.58 | 33.66 | 27.68 | 34.69 | 29.52 | 36.22 |
| **Burnt area** | area | 0.05 | 1.06 | 0.1 | 2.24 | 0 | 0.27 | 0 | 0.28 | 0.12 | 2.28 | 0 | 0.05 | 0 | 0.06 | 0 | 0.12 |
| | relative | 0.2 | 3.16 | 0.63 | 7.9 | 0.01 | 0.49 | 0.01 | 0.77 | 0.61 | 7.07 | 0 | 0.13 | 0 | 0.16 | 0.01 | 0.82 |
| **Heat Stress** | area | 1.56 | 9.59 | 2.24 | 13.81 | 1.15 | 8.79 | 1.2 | 8.6 | 2.31 | 12.49 | 0.38 | 4.89 | 0.46 | 5.78 | 0.4 | 3.86 |
| | relative | 6.31 | 22.8 | 12.11 | 34.61 | 2.65 | 14.02 | 5 | 19.34 | 10.95 | 29.43 | 1.67 | 12.24 | 1.99 | 13.9 | 4.61 | 20.88 |
| **Wind** | area | 0.7 | 3.09 | 0.94 | 3.93 | 0.52 | 2.85 | 0.6 | 2.84 | 0.95 | 3.53 | 0.13 | 1.05 | 0.18 | 1.49 | 0.2 | 1.2 |
| | relative | 2.95 | 8.69 | 5.5 | 13.09 | 1.23 | 5.02 | 2.57 | 7.34 | 4.8 | 10.55 | 0.57 | 2.91 | 0.8 | 4 | 2.37 | 7.56 |
| **Rainfall seasonality** | area | 2.72 | 9.88 | 4.31 | 14.33 | 1.44 | 8.65 | 1.95 | 8.59 | 4.56 | 13.03 | 0.47 | 4.79 | 0.55 | 5.95 | 0.61 | 4.05 |
| | relative | 10.48 | 23.32 | 21 | 35.44 | 3.29 | 13.82 | 7.92 | 19.33 | 19.54 | 30.31 | 2.05 | 12.03 | 2.35 | 14.25 | 6.97 | 21.69 |
| **Population density** | area | 0.01 | 0.15 | 16.2 | 26.24 | 0.02 | 0.3 | 0.02 | 0.33 | 0.01 | 0.13 | 0.02 | 0.34 | 0.02 | 0.34 | 0.01 | 0.12 |
| | relative | 0.05 | 0.47 | 49.96 | 50.14 | 0.05 | 0.55 | 0.08 | 0.91 | 0.04 | 0.43 | 0.08 | 0.95 | 0.08 | 0.93 | 0.08 | 0.81 |
| **Urban area** | area | 0.35 | 0.58 | 2.35 | 4.21 | 0.66 | 1.1 | 0.64 | 1.29 | 0.3 | 0.57 | 0.69 | 1.44 | 0.74 | 1.47 | 0.25 | 0.51 |
| | relative | 1.48 | 1.76 | 12.64 | 13.89 | 1.54 | 1.99 | 2.74 | 3.47 | 1.55 | 1.87 | 3 | 3.96 | 3.15 | 3.95 | 2.91 | 3.36 |
| **Cropland area** | area | 2.33 | 3.94 | 4.48 | 9.69 | 3.43 | 5.63 | 3.1 | 5.9 | 1.95 | 3.44 | 1.84 | 3.72 | 2.01 | 3.92 | 0.88 | 1.78 |
| | relative | 9.13 | 10.81 | 21.65 | 27.07 | 7.5 | 9.45 | 11.99 | 14.12 | 9.4 | 10.31 | 7.63 | 9.59 | 8.15 | 9.87 | 9.74 | 10.83 |
| **Pasture area** | area | 4.22 | 8.78 | 6.29 | 12.62 | 4.84 | 10.03 | 3.32 | 8.32 | 5.21 | 10.94 | 2.4 | 6.75 | 2.87 | 7.58 | 1.37 | 3.76 |
| | relative | 15.38 | 21.28 | 27.92 | 32.6 | 10.27 | 15.68 | 12.76 | 18.82 | 21.69 | 26.75 | 9.72 | 16.15 | 11.24 | 17.46 | 14.36 | 20.43 |
| **Burnt area with no human Pressure** | area | 0.06 | 1.74 | 0.13 | 3.15 | 0 | 0.53 | 0 | 0.52 | 0.14 | 3.46 | 0 | 0.1 | 0 | 0.13 | 0 | 0.23 |
| | relative | 0.26 | 5.17 | 0.79 | 10.78 | 0.01 | 1.01 | 0.02 | 1.62 | 0.8 | 10.84 | 0 | 0.33 | 0 | 0.42 | 0.01 | 1.62 |

Other biomes are aggregated from Olsen et al. as per[29]. Tree cover "area" is the difference in land covered by trees with and without the control's influence. "Relative" is the relative difference (i.e. tree cover "area" relative to the tree cover unconstrained by the factor/control). 10% and 90% percentiles accounts for framework uncertainty. The colours in the table are shaded according to the intensity of impact, highlighting where the stresses or pressures have the greatest effect, with darker shades indicating larger impacts.
Deciduous vegetation shows the impact on tree cover weighted by the grid-cell fraction of deciduous vegetation from ref. 85, regridded as per ref. 78.

cover above 2000 mm/yr in areas with extreme seasonality and high temperatures. Fire impacts occur in phenologically seasonal regions (Fig. 4), but not exclusively.

Fire's impact is significantly smaller than the 3.4–34.9% ($P = 0.066$–0.0003) across all FDGVMs in the fire model intercomparison experiments (FireMIP)[6]. FireMIP compared the difference between fire-on and fire-off experiments that both ran throughout the historic period. While similar to our experiment, the historic period may have had transient fire impacts on tree cover, such as fire regime shifts and deforestation fires, not tested by our framework. The significant and substantial impact, particularly in savanna climates, which have seen consistently high levels of

burning for millions of years[42] and are therefore less likely affected by transient effects, still suggests that FDGVMs fire impact on tree cover is too strong for some regions in the present day. However, humans can substantially diminish fire impact on tree cover in savannas. In areas where land use reduces tree cover, fire can only impact tree cover over this reduced area, and land use and active suppression tend to reduce burnt area in the tropics by inhibiting fire spread even in areas nearby but outside agriculture[29,43]. We quantify fire impact on tree cover without human influence by simulating the impact on tree cover from the burnt area we would expect to observe without human modification, which we obtained from[29] (see methods) and without population density or land use influence

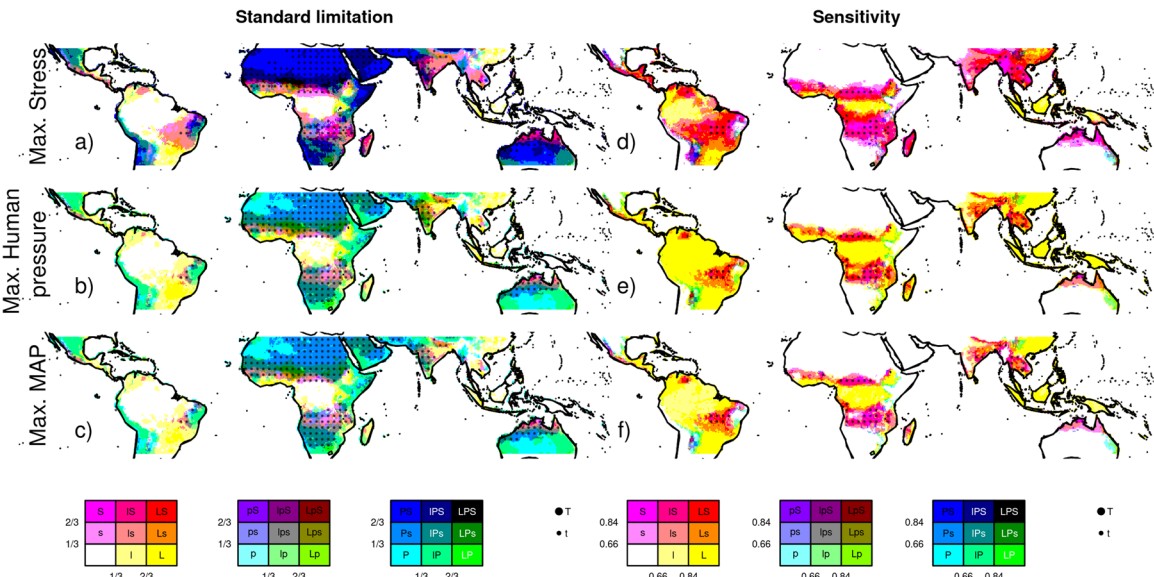

**Fig. 1 | Limiting controls on tree cover. a–c** Shows the relative standard limitation for each control and **d–f** normalised sensitivity of each factor. Purple shows areas limited by mean annual environmental stresses (S), yellow by human pressure from population density and land use (L), Cyan by Mean annual Precipitation (P) and dots by Mean Annual Temperature and Shortwave Radiation (T). Red represents co-limitation by S&L, blue by S&P, and green by L&P. Shades show the relative importance of the limitation, with darker, intense shades indicating a stronger impact, lighter shaded (and none-capitalised letter in legend) less impact, and white indicating little or no limitation - by definition coinciding with high tree cover. From top-bottom maximum stress, human pressure, and MAP limitation at 10% likelihood.

on tree cover (Fig. 5). Without direct human impact, fire would limit tree cover by 0.3–5.2% - more comparable to FireMIP models, though a similar experiment without humans has not been assessed as part of FireMIP yet. Without humans, fire's impact on savanna is 0.8–10.8% (Table 1). Although less than other stressors, it has a substantial effect, highlighting the importance of including human factors in fire and vegetation modelling.

Fire impact on tree cover in areas with high burned area is more substantial, comparable to cropland and pasture (Fig. 6). It reduces tree cover by 0.2–0.6 per unit burnt area at burnt areas >60%, compared to 0.7–1.0 per unit cropland area and 0.4–0.6 per unit pasture area (Fig. 7). However, annual burning covers a smaller area than land use - 3.8–5.7% of the tropics (depending on the dataset used, Supplementary Fig. 6). There is considerable uncertainty in the impact when burnt area is low, with tree cover reduction up to 10 times the annual average burning in areas with little fire. This suggests that, in regions with small burnt areas, the introduction of fire may still substantially impact tree cover. As burnt areas increase, their impact on tree cover is roughly linear. In contrast, the impact of heat stress increases sharply at temperatures above 35 °C and rises with wind up to 4 m/s with the possibility of much larger impacts for windspeeds >10 m/s. Relative tree cover impacts increase exponentially with rainfall seasonality, linearly to urban and cropland areas, and plateaus when pasture area cover is 20%.

Fire has little effect on tree cover bimodality (Fig. 9). No variable contributes solely to the intermediate tree gap identified in refs. 17,44 though removing the impact of heat stress or rainfall seasonality splits the gap to lower and higher tree covers (i.e. from between 33% and 98% to between <20% and ~40% and between ~60% and ~98%). Removing the effect of either cropland or pasture reduces the gap's magnitude.

**Sensitivity of tree cover to disturbance**

Tree cover in most savanna, grass and desert areas of the tropics is insensitive to small changes in any controls, i.e., for trees to establish, they require a considerable reduction in the limitation imposed by both MAP and other stresses. (Figs. 1 and 2). However, in seasonal semi-arid areas, which are susceptible to shifts in stress[45] and where rainfall seasonality and heat stress have the greatest impact (Fig. 2), marginal changes in rainfall patterns and temperature could have a major impact on tree cover (Figs. 1 and 7). South

American forests and savannas are most sensitive to changes in land use, suggesting agricultural practices could have the largest impact on future tree cover in these regions. Rainforests across the Amazon, Congo, Borneo, New Guinea Southern lowlands and Yunnan/Guizhou evergreen and dry deciduous forests are also sensitive to land use impacts. Additionally, Amazon and Indonesian rainforests show vulnerability to stress in key deforestation areas (Fig. 2).

Marginal changes in burnt area have minimal impact on savanna tree cover. Still, they could substantially impact tree cover in forest areas (Fig. 8). A 1% increase in burnt area could decrease tree cover by as much as 2% (at the model distributions 90th percentile) in the southern and eastern Amazon basin, the Amazonia arc of deforestation and South America's Atlantic forests. Smaller but important regions of sensitivity to burning also include Gabon and the western Congo forests (1.5–2% tree cover reduction with a 1% burnt area increase), southern Indonesia (1–2% reduction), the south of China moist forests (up to 2% reduction) and the eastern India dry forests (1–1.5% reduction). However, there is considerable uncertainty in this sensitivity to fire, and our framework suggests that little to no change in tree cover is also possible (Fig. 8).

**Implications and uncertainties**

Aside from MAP (mean annual precipitation), we show that rainfall seasonality and land use are likely dominant factors in maintaining tree cover at current levels. Beyond these, co-limitation from different controls (Figs. 1 and 2) suggests that no single factor influences savanna tree cover, noting that soil texture and fertility, not included here, could also determine the location of forest-savanna transitions[20] with variations at finer resolutions than tested here. All variables have a negligible effect on the bimodality of the reconstructed tree cover (Fig. 9) and are unlikely to be a cause of the "missing" intermediate tree covers at the resolution we test. Alternative drivers for missing intermediate tree cover include the natural distribution of cover expected under random fluctuations[46] and biases in tree cover observations, which tend to underestimate tree cover in savannas[21,24]. Our results suggest that either of these or factors not included, such as soils, are more likely drivers of tree cover bimodal distribution. As relevant soil properties vary at a finer resolution than there are detailed tropical observations, we did not test soil here.

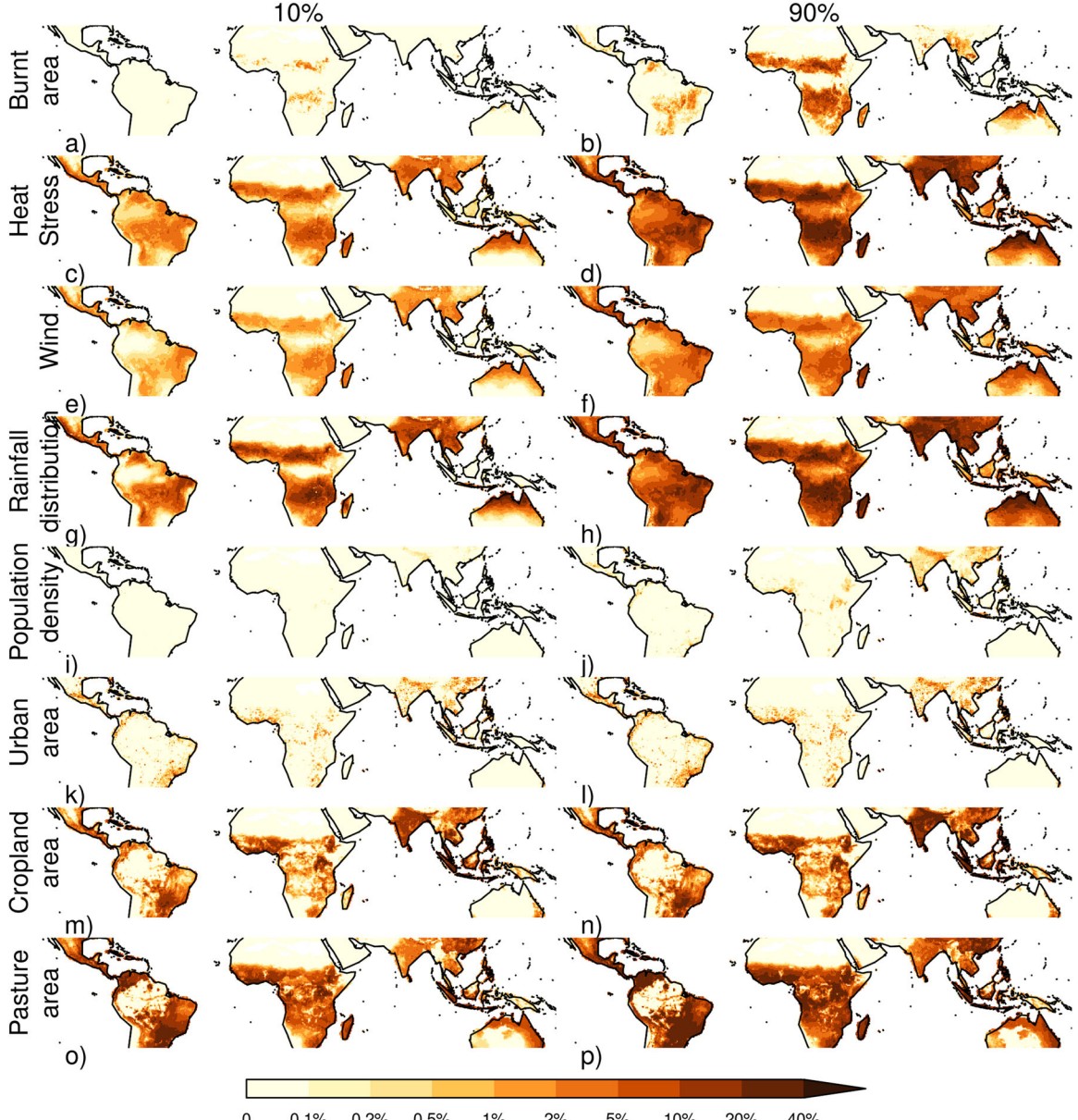

**Fig. 2 | The percentage reduction in tree cover area by each environmental and human stresses.** Each row represents a different stress (top-bottom): **a, b** Fire (using burnt area), **c, d** heat stress, **e, f** windthrow, and **g, h** seasonal rainfall distribution. These are followed by human pressures: **i, j** population density, **k, l** urban area, **m, n** cropland area and **o, p** pasture area. For each stress or human pressure, two maps are shown: the left map represents the 10th percentile, and the right map represents the 90th percentile of the likely range of the stress' impacts, illustrating the range of uncertainty in the estimated tree cover reduction. This figure allows for a visual comparison of both the magnitude of tree cover reduction by each stress and the confidence level (percentile range) associated with these reductions.

However, future work in specific locations and finer scales could incorporate soil properties.

The impact of land use on tree cover does not always match the extent of the tree cover itself. This is because, in addition to the extent of land use changes, reductions in tree cover may diverge due to various factors, including spatial heterogeneity, differential sensitivity of vegetation types, ecological resilience and regeneration processes, fragmentation effects, management practices, and climate and environmental factors. While some of these impacts extended beyond the land cover extent itself, we safely assumed that they occur within the same gridcell, given the coarse scale of analysis (0.5°, ~50 km) employed in this study. That tree cover responses follow cropland extent (Fig. 6) suggests that any additional impact on tree cover beyond cropland extent are negligible on our coarse scales. Urban areas do have a large impact beyond their extent – up to 10 times at lower urban covers,

possibly owing to factors such as heat island effects, altered microclimates, fragmentation of surrounding ecosystems, and direct human disturbances such as deforestation and land clearing for urban expansion. Pasture, however, shows a smaller impact than pastures own extent, especially as pasture area increases, indicating high tree cover retention at higher pasture areas.

Heat stress and windthrow have a substantial impact on tree cover which is in line with[47,48]. Few DGVMs incorporate direct effects of either (though note[8]), which our results suggest might also aid simulated vegetation distribution. For example, heat stress affects the productivity of vegetation. It can have large implications for the resilience of tropical forests towards the more severe climate projections into the twenty-first century[49]. While DGVMs represent declines in productivity at higher temperatures (and lower precipitation), some mortality mechanisms, such as xylem embolism during extreme heat and drought[50], may be underrepresented in

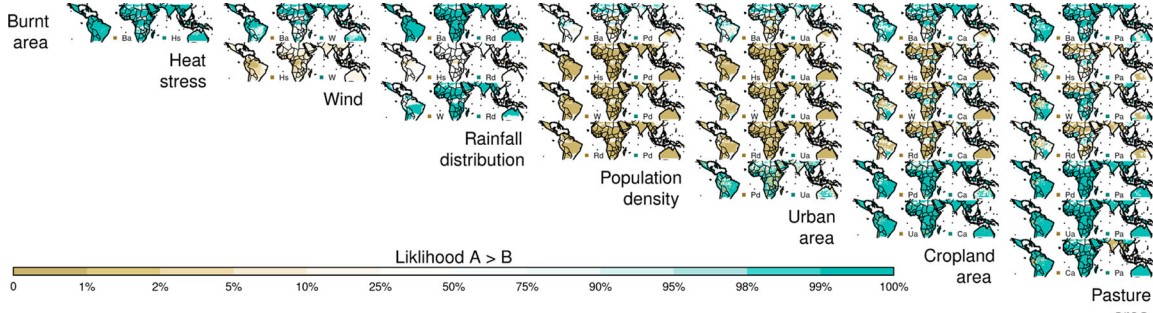

**Fig. 3 | Pairwise comparison of the likelihood that the stress or human pressure in each column reduces tree cover more than the one in each row.** Each map in the grid shows indicates the likelihood of the column stress having a greater impact on tree cover reduction than the row stress. Blue areas represent regions where the column stress is more likely to cause a higher reduction in tree cover, while brown areas represent regions where the row stress has a higher likelihood. The stress or pressure's first two letters or initials are listed next to the relevant colour for each map. For example, the top left blue areas show where Hs (Heat stress) reduces tree cover more than Ba (Burnt area). White indicates equal likelihood, and lighter shades of blue or brown show a slight likelihood difference between the column and row stress. The colour gradients allow for a visual comparison of how different stresses or pressures are likely to impact tree cover in various locations.

models[51]. Also biotic attacks following drought and/or windthrow events[52] are generally not represented in models.

Fire has a surprisingly low influence on tropics-wide tree cover, though it plays a more important role in savannas, suppressing tree cover by 0.6–7.1%. Without human impact on burnt area and tree cover loss, the impact on fire in savannas has the potential to be much higher at between 0.8 and 10.8% (Table 1). This is more in line with empirical studies and field experiments[13,53] though it still shows that the independent impact of fire is not enough to fully explain the lack of tree cover within savannas. The use of remotely sensed data may contribute to the surprisingly low impact. Overstory mortality is generally minimal in frequently burned woodlands and savannas, with frequent surface fires primarily influencing recruitment through high seedling/sapling mortality. Therefore, the observed low impact of burnt areas from surface fires on tree cover, as detected by satellites, may reflect the resilience of mature trees to fire-driven mortality in these environments. However, it is worth noting that FDGVMs tend to target remote sensed burnt area for parameterisation and evaluation[11,32,54].

We were able to separate out the effects of different co-varying impacts – which is challenging in many field-based and empirical studies that often consider fire in isolation from other dry disturbances[13]. These findings are not inconsistent with the idea that fire has an important impact on vegetation cover and species selection but suggest that tree covers are mostly at equilibrium in fiery landscapes with present-day fire regimes.

By considering the controls on the static distribution of tree cover, the study does not look at the sensitivity of factors that may determine forest cover temporarily. In deforestation hotspots, increased deforestation rates are associated with increased burning[30], which can then decrease due to subsequent land fragmentation[35] – a process not inconsistent with but not tested in this study. Large-scale tropical deforestation also leads to warmer, drier and more seasonal conditions[55,56] that our results suggest may maintain low tree cover. Fire may also play a role in future forest transitions, resulting from interactions between land use change, fire and land-atmosphere exchange in a changing climate.

The slight reduction in tree cover could be because of coarse (0.5° × 0.5°) spatial scale. Other studies show a substantial impact (up to 20%[13]) of fire on tree cover at fire return intervals of around 1–10 years, which would result in coarse-scale burnt areas of (1/return time) 10–100% burnt area[53]. Here, we show tree cover is reduced only slightly less than the annual average area burnt – consistent with these finer-scale studies and on par with the impact of agricultural land use per unit area (Fig. 7). Therefore, our results do not preclude a substantial impact of fire on cover if the same areas within a given grid cell are burning each year, which would also explain tree cover changes found in fire exclusion experiments[57]. In mesic tropical systems (savanna and dry forest), fire effects on tree cover can be substantial if vegetation experiences frequent fires, particularly later in the dry season[19]. These regions show a higher impact (Table 1; Fig. 4). However, burning here

still only impacts relative tree cover by up to 20%, with many areas seeing little impact. Tree species in ecosystems prone to regular fires demonstrate adaptations that allow them to survive, resprout and recruit in the presence of burning[58,59]. These effects could be tested by representing subgrid heterogeneity or applying this framework at finer resolutions. From an Earth System perspective, fire and the other stressors tested do not just impact vegetation distribution. There are also disturbance-driven variations in other important wood vegetation processes such as height, carbon uptake, carbon allocation, hydrology, and ecosystem fluxes[6,60,61], all of which are influenced by fire.

Fire impact on tree cover is less than previously found in DGVM studies[6]. Many fire-enabled DGVMs incorporate the impact of precipitation through carbon dynamics, which determines vegetation distribution to some extent. However, most models also require a substantial impact from fire to simulate the correct distribution of tree cover. As fire impacts co-vary with other stress factors, most notably seasonal rainfall distribution (Supplementary Fig. 7, Supplementary Fig. 8), DGVMs might overly rely on fire to simulate correct tree cover distribution because they underestimate vegetation response to moisture availability. Some FDGVMs use fire intensity to describe fire impacts on cover[11,54], and incorporating intensity into this framework may help constrain the broad uncertainty in fire impact - particularly in regions with low burnt areas and high tree covers (Fig. 7). However, despite the wide distribution of potential fire impacts, even the most extreme of our framework's posterior probability distribution suggests that fire reduces tropical tree cover significantly less than any tested FDGVM. This assumes that switching off fire from our present-day fire-on state would result in vegetation cover equal to the fire-off simulation. Both simulations have been spun up with and without fire, respectively, which may lead to different stable states in the present day. More targeted fire model experiments (e.g. switching off fire at the present day and running to equilibrium) are needed to provide a more specific test. Adapting our framework to test the transient impacts of fire on cover (from, e.g. deforestation fires) will also help attribute the differing impact fire can have on tree cover over time.

We took advantage of the work in FireMIP to compare how fire modulates tree cover to our observational constraints. Other model inter-comparisons could perform similar factorial experiments to compare to our results. The constraints we have found on the impact of wind and heat stress could be particularly useful to assess and re-parameterise vegetation models that represent both these disturbances. This might also establish if the larger impact of fire is due to little impact of climate seasonal rainfall, windthrow or heat stress. Like DGVMs, our model considers similar responses of tree cover across continents with different evolutionary histories and across gradients of anthropogenic landscape modification. Further framework development could attribute uncertainties in tree cover disturbance

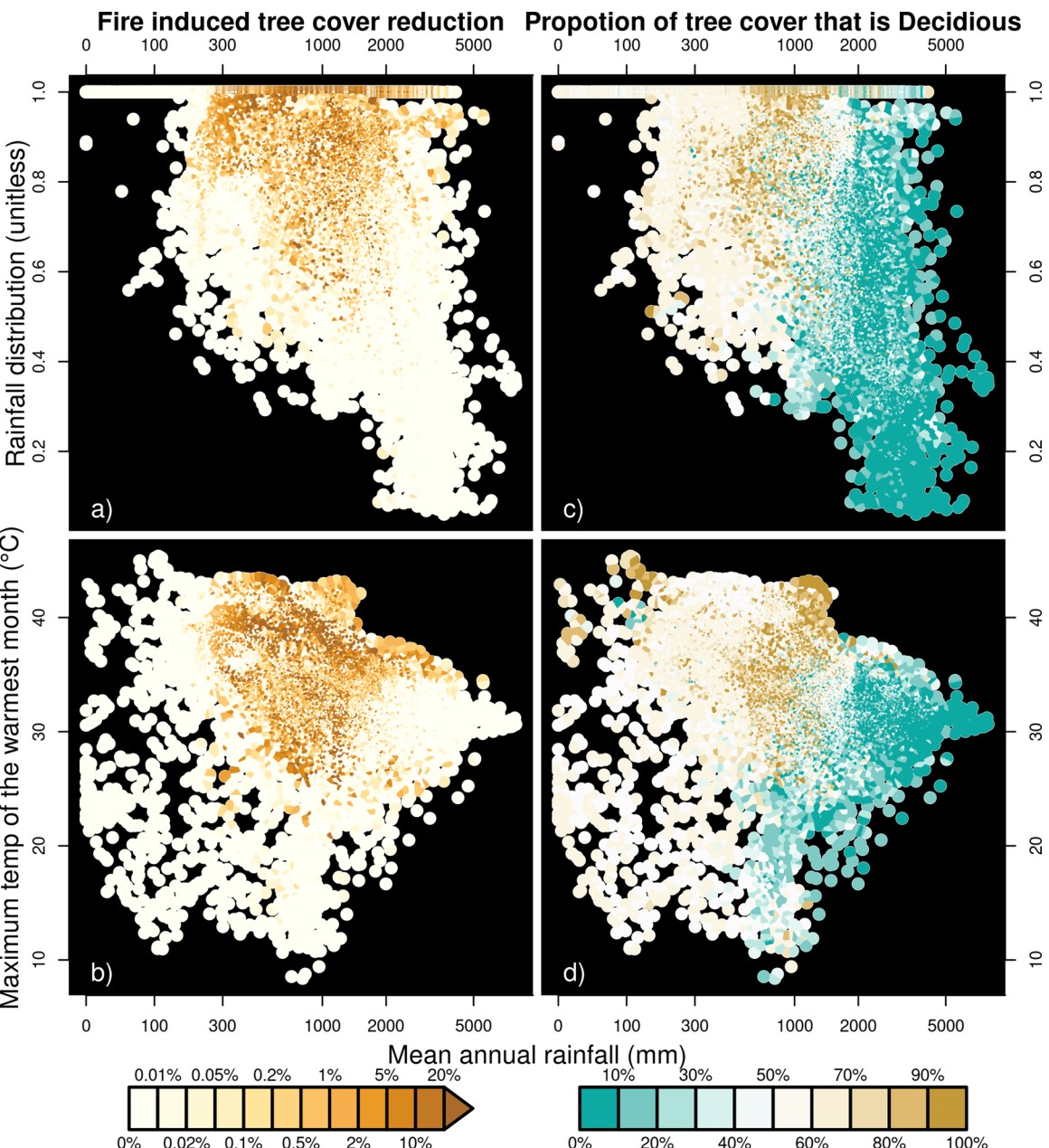

**Fig. 4 | The impact of burnt area on tree cover in bioclimate space.** Dots indicate grid cells with (x-axis) Mean annual rainfall and (y-axis) rainfall seasonality for **a**, **c** and maximum temperature of the warmest month for **b**, **d**. **a**, **b** Colour indicated fire impact on tree extent and **c**, **d** percentage deciduous cover vs. evergreen (see methods).

responses to these assumptions, which may also help guide DGVM development priorities.

Our framework shows considerable uncertainty in the impact of fire in humid forests, which, historically, have not experienced regular burning. Our framework suggests a plausible range from virtually no sensitivity of forests to burning up to more than twice as much forest loss as changes in burnt area. However, as these areas have not historically experienced high fire occurrence, their vegetation is unlikely to be fire-adapted. And many studies have highlighted tropical forest areas that experience infrequent burning as particularly sensitive to even small increases in burning[62–64]. We have not considered the distribution of fire resilience or acclimation to fire in this study. There has been a considerable shift in burnt area controls found in many of the pantropical forest areas[29], and a recent UN report highlighted Eastern and Southern Amazon and Indonesia as at risk for substantial increases in wildfire occurrence[35]. In these regions, tree cover could still be significantly affected by fire under future environmental change, particularly if the speed of such a change precludes the establishment of more

adapted tree communities typical of more fire-prone wooded ecosystems[65–68].

## Conclusion

Our results have shown that, over most of the tropics, human impacts, rainfall seasonality, windthrow, and heat stress have a much more substantial direct impact on tropical tree cover than fire. Simulation of tropical forest and savanna distribution should focus more on tree cover responses to moisture and temperature. More emphasis on simulating non-fire disturbance events may also improve simulated tropical vegetation distributions. However, we show that tropical forests are potentially sensitive to small amounts of burning, even without considering the heightened vulnerability of these forests due to their lack of fire history. Given the increased trend in burnt area in these regions[29,35], fire could seriously reduce tree cover under future climates, resulting in modification of important climatic feedbacks that would fundamentally change the carbon dynamic in the tropics. Therefore, based on our results, we recommend targeting FDGVM

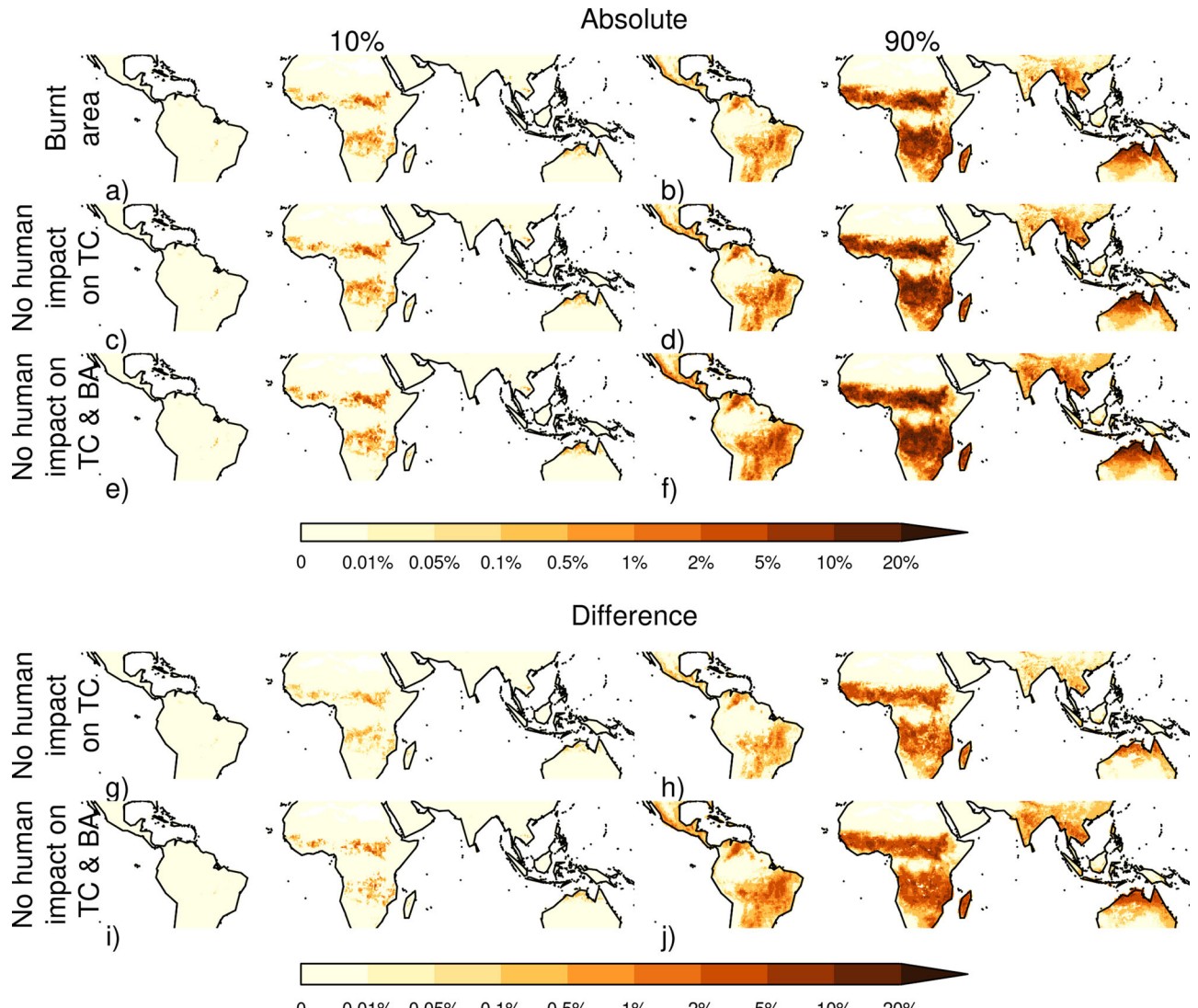

**Fig. 5 | The percentage reduction in tree cover area by fire. a, b** Fire as per Fig. 2, **c, d** fire without direct human influence on tree cover and **e, f** fire without influence from human impact on tree cover or burnt area. **g, h** Shows the added impact on tree cover fire would have without humans influencing tree cover, while **i, j** shows fires added impact without human influence on tree cover or burnt area. Columns show 10% and 90% percentiles accounting for framework uncertainty.

development to simulate fire and vegetation responses to burning in tropical forested areas - essential for evaluating future impacts of fire in these carbon-rich forests.

## Methods
### Modelling framework
We constructed a Bayesian limitation control model to estimate probability distributions describing the likelihood of percentages of tropical tree cover based on a set of environmental predictors similar to the framework outlined in refs. 29,30. These predictors, derived from spatial maps, were utilized as inputs to the model, which was optimized to generate tree cover maps as outputs.

In the framework, the 11 predictors influencing tree cover are linearly combined into limiting controls (Supplementary Fig. 1; Supplementary Table 2): (i) mean annual precipitation ($MAP_*$); (ii) mean annual temperature ($MAT_*$); (iii) shortwave radiation (combining diffuse and direct radiation sources) ($SW$); (iv) environmental "stress" ($S$) from fire, rainfall seasonality, heat and windthrow; and (v) human pressure (comprised mainly of "land use", dubbed $LU$) combining urban, cropland, pasture and population density. (ii) and (iii) are multiplied together to form an "Energy"

limiting factor when displayed in the results. $MAP_*$, $MAT_*$ and $SW$ climate controls are often used to understand vegetation distributions[1,69] and have been used by limitation studies exploring controls on net primary production[70]. Grouping the stresses into one control follows the concept that woody plant resilience and recovery strategies (e.g. resprouting) help plants avoid mortality and rapidly re-establish after a range of stresses[61].

Each control was expressed as a linear combination of its respective factors. Fractional tree cover ($TC$) was calculated as a product of limitations imposed by control ($f(k_c \times (X_c - X_{0,c}))$ where $c$ is a control ($\mathbb{C} = \{MAP_*, MAT_*, SW, S, LU\}$), with each control's limitation represented by a logistic curve, $f$:

$$TC = TC_{\max} \times \prod_c^{\mathbb{C}} f(k_c \times (X_c - X_{0,c}))$$

$$f(x) = 1/(1 + e^{-x}) \tag{1}$$

Where $TC_{max}$ is used to aid our model optimisation as per[29,30]. $X_{0,c}$ is the value of $X_c$ which reduces tree cover to 50% of its unconstrained area,

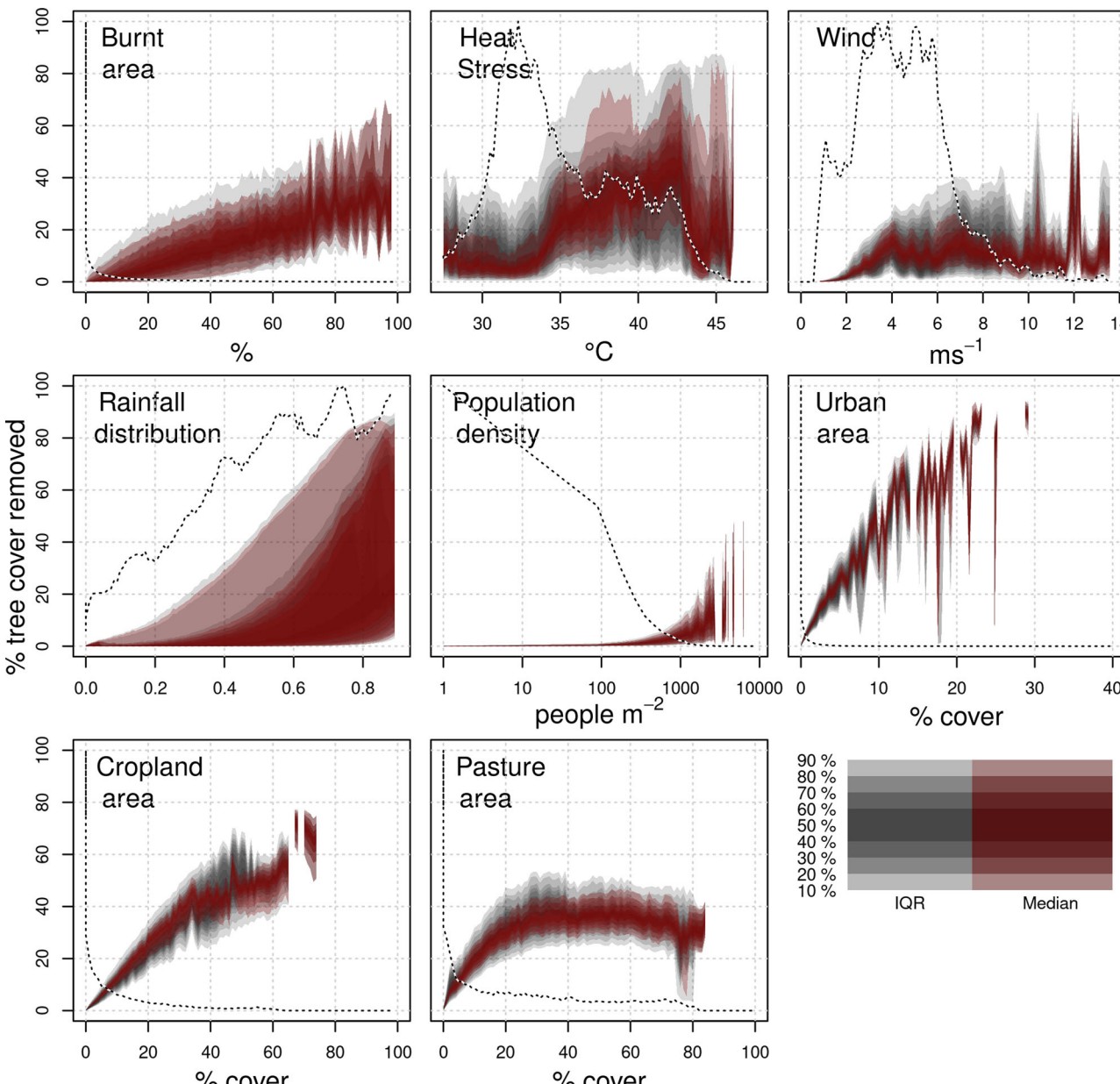

**Fig. 6 | The relative response of % tree cover to each environmental or anthropogenic stressors.** calculated as the difference in tree cover with and without each stressors divided by tree cover without that control (Eq. (8), methods). Grey areas represent the 10–90% uncertainty percentiles (in 10% increments) of the response interquartile range. Red areas represent the 10–90% percentiles (in 10% increments) of median response. The dotted lines show the frequency of occurrence of controls.

and $k_c$ is the steepness of the logistic curve, equal to ¼ of the gradient at $X_c = X_{0,c}$. $k_c > 0$ where tree cover increases with the control (i.e. $MAP_*$, $MAT_*$, $SW$), while $k_c < 0$ and tree cover decreases for suppressive controls ($S$ and $LU$).

We assume that there is no tree cover when there is no rainfall and, therefore, perform a log transform on $MAP$ (i.e. $MAP_* = \log(MAP + 1/ncells)$ where ncells is the number of grid points), making our control curve (Eq. (1)) tend to zero as $MAP$ tends to zero. As $MAT$ and $SW$ had little impact individually on tropical tree cover, we combined both into energy ($E$) control were $f_E = f(MAT_*) \times f(SW)$.

We represented $S$ and $LU$ controls by combining factors ($x_i$) weighted by their respective influence ($v_i$). For $S$ control, as we describe cumulative effects of annual average stresses, we do not assume that stress impact accumulates linearly with each increased stress. For example, an increase in one-degree temperature stress will likely have a much bigger impact at high temperatures. To account for this, we raise each variable to a power. Therefore:

$$X_c = \sum_i v_i \times x_i^{p_i} \bigg/ \sum_i v_i \text{ where } v_1 = 1 \text{ and for } f = SW, LU, p_i = 1$$

(2)

## Datasets

We optimised the framework against MODIS Vegetation Continuous Fields (VCF) collection 6 fractional tree cover[31], regridded as per ref. [29]. Although refs. [21,24] recently demonstrated a potential bias in the tree cover distribution of VCF, there are no alternative global fractional tree cover datasets available which are independent of VCF, so our interpretation of results focuses on contrasting the impact of different controls, which, given our model setup, only assumes that VCF values regridded to 0.5° are correctly ranked.

**Fig. 7 | Response curves of the relative impact on tree cover per unit of burnt area (top left) or land use (other panels) in 1% burnt area/land use bins.** Calculated as $(TC_*(i) - TC)/(A_{bin} \times TC_*(i))$ where $A_{bin}$ is the burnt/land-use area of each bin and $TC_*(i)$ is tree cover without burnt area/land use (i.e. a value of 1 indicates a 100% tree cover exclusion from the burnt/land-use area). Dotted lines show the frequency of occurrence of burnt area/ land use.

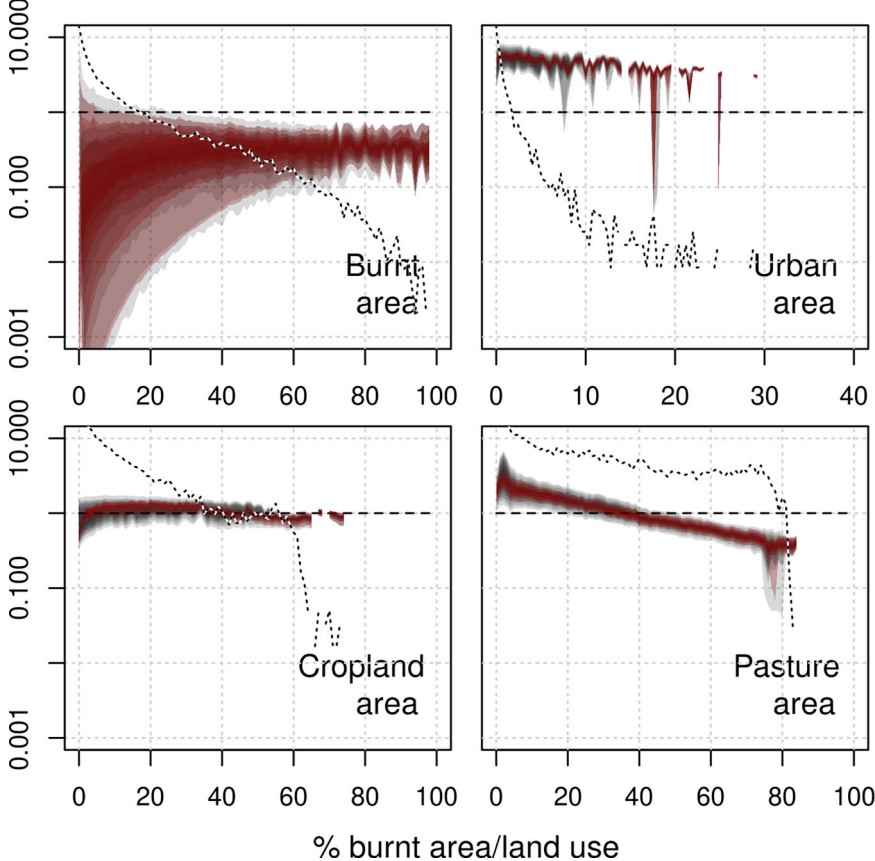

Disagreement between burnt area datasets can significantly affect analyses of fire and vegetation interactions[71]. We, therefore, ran the framework using five different burnt area datasets used in and provided by the Fire Model Intercomparison Project[11,32,54] (Supplementary Fig. 6). Likewise, to sample the well-known disagreement between precipitation datasets[72], we used four different precipitation products (Supplementary Fig. 8): GPCC[73]; MSWEP[74] and CMORPH[75,76] downloaded from the eartH2Observe portal (http://www.earth2observe.eu/); and CRUTS4.03[77] downloaded from CEDA.

A variety of metrics can be used to describe rainfall seasonality. As there is no single apparent metric candidate, we ran separate optimisations with the following metrics (Supplementary Fig. 8):

- Fractional mean annual dry days (*MADD*), calculated as 1 - fractional number of wet days for CRU, and number of days where rainfall is less than 0.1 mm for all other products.
- The fractional number of dry days in the driest month (*MDDM*) - as above, but for the month with the smallest number of dry days.
- Precipitation in the driest month (*MADM*) - which we normalised by the mean monthly precipitation, i.e.:

$$MADM = \min\{pr_m\} \times \frac{12}{MAP} \qquad (3)$$

Where $pr_m$ is the monthly precipitation climatology over our study period.

- The mean seasonal concentration of precipitation (*MConc*) is calculated as per the concentration metric[78]. i.e.:

$$Mconc = \frac{\sqrt{L_x^2 + L_y^2} :}{\sum_m pr_m}$$

$$\text{where } L_x = \sum_m pr_m \times \cos(\theta_m) \text{ and } L_y = \sum_m pr_m \times \sin(\theta_m) \qquad (4)$$

Where $\theta_m$ corresponds to the direction of the month, *m*, in the complex plane.

We calculated annual averages between 2000 and 2013 - the dataset's common period - for the region between 30° North and 30° South to delimit the tropics and subtropics at 0.5° resolution - the dataset's most common resolution. We performed bilinear resampling for data not already on a 0.5° grid, using the R Package 'raster'[79].

**Optimisation**

We optimised our framework's reproduction of tree cover using a Bayesian inference technique[41]. Bayesian inference allows us to quantify framework uncertainty, including uncertainty for variables with multiple realisations, and therefore provides confidence in our driver's impacts - applicable when assessing co-varying variables. Instead of giving us a single, defined set of parameters, Bayesian inference produces the probability distribution for each parameter, which we propagate through to the influence of controls, limitation factors and overall tree cover. Bayes theorem states that the likelihood of the values of the unexplained parameter set, $\beta$ (i.e. parameters in Eqs. (1)–(4)), as well as our error terms ($\sigma$, $P_0$), given a set of observations (*Obs*), is proportional to the prior probability distribution of $\beta$ ($P(\beta)$) by the probability of *Obs* given $\beta$. i.e

$$P(\beta|Obs) \propto P(\beta) \times P(Obs|\beta) \text{ where}$$

$$\beta = \{\{X_0\}, \{k\}, \{v_i\}.\{p_i\}, TC_{max}.\sigma, P_0\} \qquad (5)$$

This gives us 23 parameters to optimise. We assume no prior knowledge and so set generously large priors on all parameters: uniform priors with only physically plausible bounds (i.e. [0, 1]) for $TC_{max}$, exponential

**Fig. 8 | Senstivity of tree cover to changes in burnt area.** Reduction in tree cover due to a 1% increase in burnt area for (top) 90% and (bottom) 10% uncertainty range. Note that the colour scales of the two maps differ.

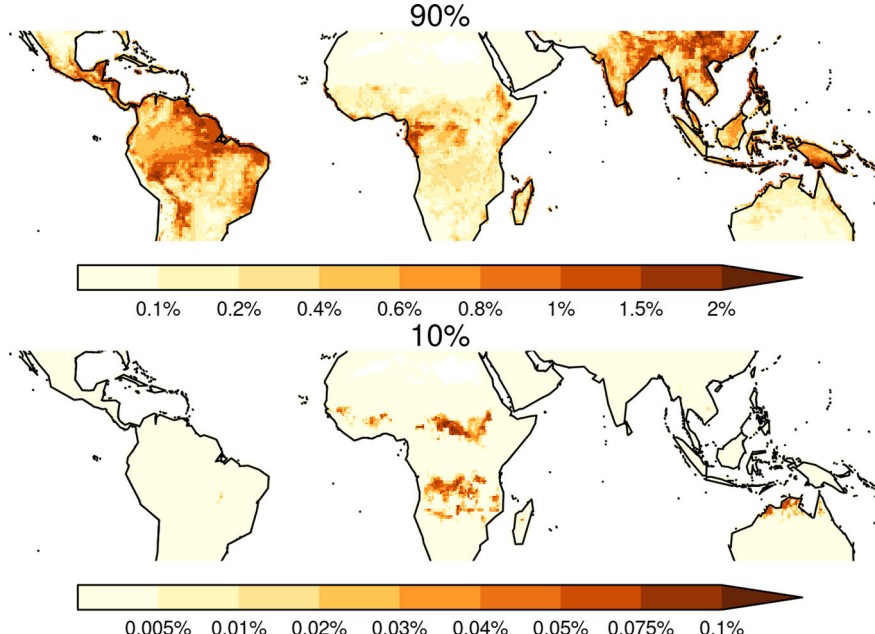

distributions with a generous rate parameter of 1 for $v_i$, $p_i$h and $k_i$ which has a lower (or upper for $k_S$ and $k_{ex}$) bound of 0, and normally distributed priors with a mean and deviation of half the range of the corresponding control (i.e. 0.5 for fractional cover for land use) for $x_0$.

Tree cover is approximately normally distributed under a logit transformation, apart from a slight divergence at the intermediate tree cover gap and a peak at 0 tree cover (Supplementary Fig. 9). We use the same zero-inflated, logit-normal distribution from ref. 30.

$$P(Obs = 0|\beta) = \prod_i^N (1 - TC_i^2) \times (1 - P_0)$$

$$P(Obs = 0|\beta) = [1 - P(Obs = 0)|\beta)] \times \frac{N}{\sigma \times \sqrt{2 \times \pi}} \times e^{\sum_i^N \left[\frac{logit(obs_i) - logit(TC_i)}{\sigma_i}\right]^2}$$

(6)

where $i$ represents an individual datapoint, $Obs = \{obs_i\}$ is our set of target observations and $N$ is the observation sample size. Inferring the posterior solution is a case of maximising Eq. (6). Inference and posterior sampling were based on the ConFire model code[30,80]. The posterior solution was inferred using a Metropolis-Hastings Markov Chain Monte Carlo (MCMC) step with the PyMC3 Python package[81]. We ran 10 chains with 10,000 iterations over 20% of the data points (i.e. $N = 2408$ points) separately for each rainfall seasonality metric and each precipitation and burnt area dataset. Unless otherwise stated, the posterior solution is constructed by sampling 10 parameter ensemble members from the last 5000 iterations of each chain (i.e, 100 samples for each dataset/seasonaility metric combination). A combined posterior was calculated by bootstrapping 1000 ensemble members across datasets and rainfall distribution metrics members, with the selection probability derived from Eq. (6). Sampling was performed using the Iris package[82] with Python version 3 (Python Software Foundation, https://www. python.org/).

**Measures of impact on tree cover**

We follow a modified version of Kelley et al.[29] definitions of limitation and sensitivity to controls. This approach allowed us to quantify both absolute and relative contributions of individual controls and factors. The impact on tree cover of a factor or control, $i$, is the absolute difference in tree cover with ($TC$) and without ($TC_*(i)$) that factors influence ($|TC_*(i) - TC|$). We used this for Table 1 and Figs. 2 and 5.

The relative impact on tree cover ($pe(i)$) of a given control or factor, $i$, is the increase in tree cover if the limitation imposed by $i$ is removed in the presence of other controls:

$$pe(i) = \frac{TC_*(i) - TC}{TC_*(i)}$$

(7)

As used in Figs. 4 and 6. For controls, $TC_*(X_c)$ is simply the product of all factors, excluding that control, ($\complement \backslash c$) i.e:

$$TC_*(X_c) = TC_{max} \times \sum_j^{\complement \backslash c} f(k_j \times (X_j - X_{0,j})) \text{ and therefore}$$

$$pe(i) = TC \times (1 - f(k_i \times (X_i - X_{0,i})))$$

(8)

If $i$ is a factor within control $c$ then we simply remove $i$ from that factor ($I \backslash i$):

$$TC_*(X_c) = f\left(k_c \times \left(\sum_j^{I \backslash i} v_j \times x_j - v_{0,j}\right)\right) \times \prod_j^{\complement \backslash c} f(k_j \times (x_j - x_{0,j}))$$

and therefore

$$pe(i) = TC \times \left(1 - \frac{f\left(k_c \times \left(\sum_j^{I i} v_j \times x_j - v_{0,j}\right)\right)}{f(k_c \times (x_c - x_{0,c}))}\right)$$

(9)

which is the product of all factors with the contribution of control $i$ within its factor, $l$ removed.

The probability of $pe$ for $i$ being significantly different than $j$ ($P(pe(i) \neq pe(j))$) is the root of the multiplied distributions, as per[35].

$$P(pe(i) \neq pe(j)) = \sqrt{\int_0^1 P(TC|\beta_{-i}|Obs) \times P(TC|\beta_{-j}|Obs)dTC}$$

(10)

Where $\beta_{-i}$ is the set of parameters with $v_i$ set to zero, removing the influence of factor (or factors if testing a control) $i$.

Reported $P$-values in the main text are then $1 - P(pe(i) \neq pe(j))$, to conform to the convention of testing for a null result.

**Fig. 9 | Frequency of occurrence of tree cover in 1% bins.** Top left for VCF observations, with colour indicating the frequency of occurrence by aggregated Olson biomes[84] (see methods). Top right for the framework with black showing 10% and grey showing 90% percentile based on parameter uncertainty and dashed line the VCF observations. This corresponds to red-shaded regions' in subsequent plots, which show tree covers from the framework when each listed environmental stress or human pressure control is removed.

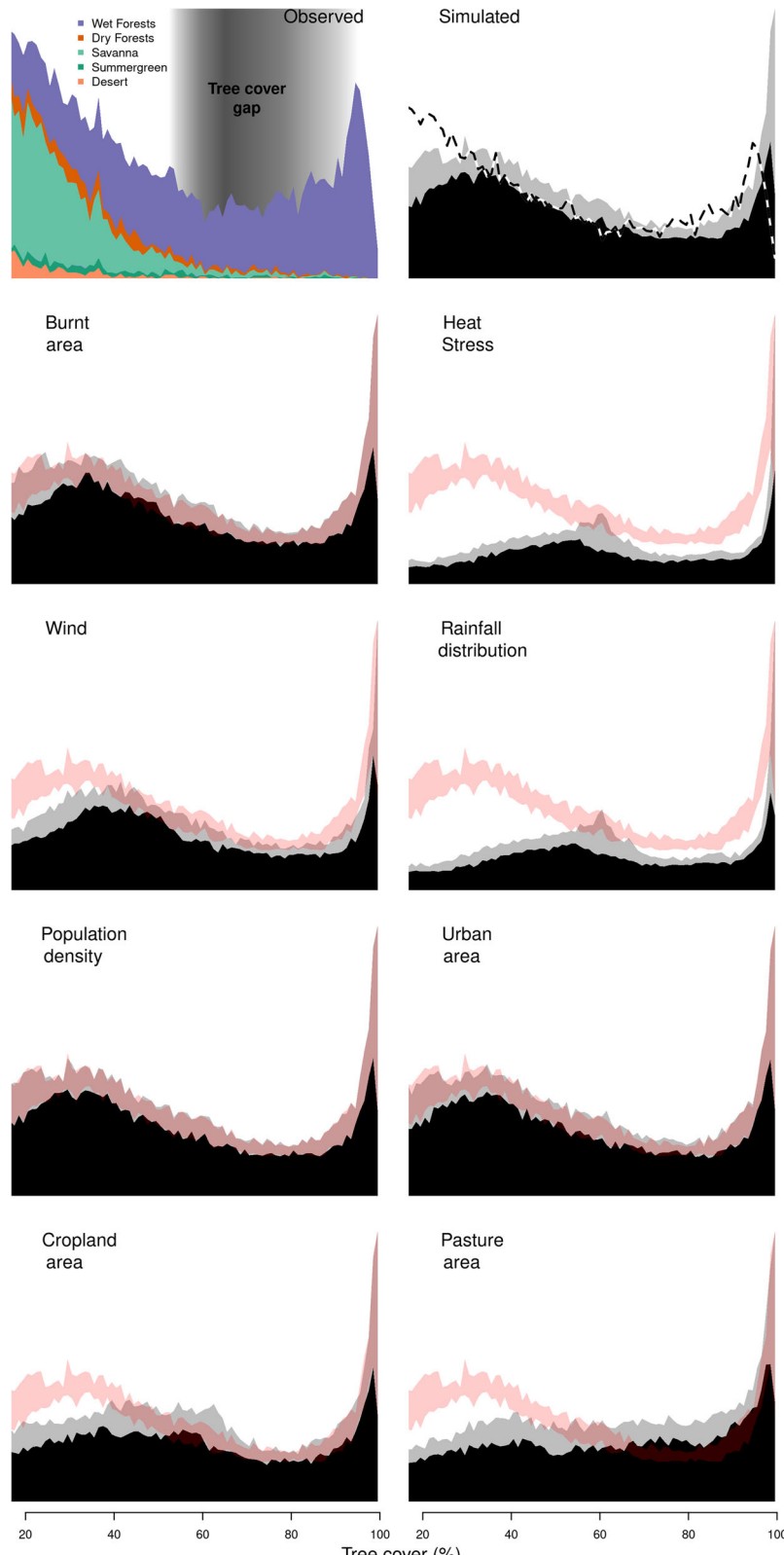

To test the impact and relative impact of burnt area in the absence of human pressure, we performed two additional tests:

1. The impact fire would have without direct human impact on tree cover. To do this, we compared tree cover without $LU$ with tree cover without $LU$ and burnt area. ie $|TC_*(burnt\ area, LU) - TC_*(burnt\ area)|$ for

tree cover impact and

$$\frac{TC_*(burnt\ area, LU) - TC_*(burnt\ area)}{TC_*(burnt\ area, LU)}$$ for relative tree cover impact

$$(11)$$

2. The impact fire would have without direct human impact on tree cover and burnt area. We used simulation from just GFED4s, which we replaced with reconstructed burnt area from the 1000 ensemble members without human influence (crop, pasture and population density) in ref. 29. Each member was randomly sampled when constructing $TC_*(burnt\ area, LU)$ and $TC_*(burnt\ area)$ in Eq. (12).

The limitation imposed by a control ($L(X_c)$) is simply one minus the maximum tree cover allowed for that factor. For $S$ and $LU$, we first normalise the maximum allowed tree cover by the tree cover when the factor is 0, i.e. so that tree cover is not limited by stress when there is no stress, or human pressure when there is no human pressure:

$$L(X_c) = \begin{cases} 1 - f\left(k_c \times (X_c - X_{0,c})\right) & X_c = MAP_*, MAT_*, SW \\ 1 - \dfrac{f(k_c \times (X_c - X_{0,c}))}{f(-k_c \times X_{0,c})} & X_c = S, Ex \end{cases} \quad (12)$$

We also evaluate how sensitive tree cover is to changes in each control to establish the resilience of the tree cover to environmental change ("sensitivity" from ref. 29). The sensitivity of a control or factor ($R(i)$) is the tree covers rate of change ($G(i)$) relative to the maximum rate of change in tree cover for that control (i.e. when $X = X_0$), again in the presence of the other control:

$$G(i) = \frac{\delta f(i)/\delta(i)}{\delta G(i)/\delta(i)} \text{ and}$$

$$R(i) = G(i) \times TC_*(i) \quad (13)$$

To test the sensitivity of tree cover to small changes in fire, we ran the framework with an increase of 1% burnt area across the tropics. The impact is simply the difference to the "standard" run with observed burnt area.

Probability densities for $L(X_c)$, $P(x)$, $R(i)$ and the fire sensitivity test were constructed using the same bootstrapping protocol described under "optimisation" as per ref. 29. The difference between the standard run was calculated by pairing ensemble members to account for co-variation amongst parameter distributions. Uncertainty estimates for limitation and sensitivity are based on our sampled posterior solution's 10% and 90% quantile range. Uncertainties in Fig. 1 estimates the 10–90% range by using the 65% quantile for the limitation or sensitivity imposed by the maximum control and the 35% quantile for all other controls.

**Biomes.** We used the Olson biome[83,84] groupings from ref. 29. Wet forests are defined as tropical & sub-tropical wet broadleaf forest, tropical and subtropical coniferous forests; Dry forest as tropical and sub-tropical broadleaf dry forest; Savanna/grassland as tropical and subtropical grasslands, savannas and shrublands, wooded grasslands & savannas; Mediterranean as mediterranean forests, woodlands and scrub; Summergreem forests/woodland as temperate broadleaf and mixed forests, temperate grasslands, savannas & shrublands, temperate conifer forests. Table 1 shows the grid-cell area weighted sum of each variable's impact and relative impact on tree cover. For "Deciduous vegetation", we based the area weights on-grid areas multiplied by the fraction of deciduous vegetation for that grid cell, where the deciduous fraction came from ref. 85.

## Data availability
Model inputs and outputs are available at https://doi.org/10.5281/zenodo.8322912[86].

## Code availability
Code for running the framework is available, and generating data used in this analysis is available at github.com/douglask3/savanna_fire_feedback_test/tree/Paper1[87] (https://doi.org/10.5281/zenodo.5513895).

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

## Acknowledgements
The contribution by D.I.K. and R.J.E. was supported by the U.K. Natural Environment Research Council through The U.K. Earth System Modelling Project (UKESM, Grant No. NE/N017951/1). G.L. was supported by the U.K. Natural Environment Research Council through the ForeSight project (NE/S010041/1). D.I.K. was supported by the Natural Environment Research Council as part of the LTSM2 TerraFIRMA project and NC-International programme [NE/X006247/1] delivering National Capability. N.D. was supported by European Research Council (ERC) funding under the European Union's Horizon 2020 research and innovation programme (grant agreement No: 787203 REALM). G.P.W. was supported by the Met Office Hadley Centre Climate Programme funded by BEIS and Defra. C.B. was funded by the Met Office Climate Science for Service Partnership (CSSP) Brazil project which is supported by the Department for Science, Innovation & Technology (DSIT). A.A. was supported by the Met Office Hadley Centre Climate Programme funded by DSIT. Analysis in this paper benefited from discussions with the JULES mortality process evaluation group (https://jules.jchmr.org), whom we would like to thank for their advice and support.

## Author contributions
Douglas I. Kelley, France Gerard, Ning Dong and Elmar Veendendaal developed the concept. Douglas I. Kelley and France Gerard designed the limitation framework methodology with advice on limitation factors from Graham P. Weedon, Richard J. Ellis, Eddy Roberston, Douglas I. Kelley, Rhys Whitley, Chantelle Burton, Ning Dong, Ioannis Bistinas, and Toby R. Marthews designed the Bayesian Inference methodology. Douglas I. Kelley, Guangqi Li and Ning Dong obtained and processed the data used. Douglas I. Kelley drafted the manuscript with help from France Gerard, Ning Dong, Elmar Veendendaal, Arthur Argles and Gitta Lasslop. All authors reviewed and edited the final draft.

## Competing interests
The authors declare no competing interests.
