## [Peer Review file · Communications Earth & Environment]

Observational constraints of fire, environmental and anthropogenic on pantropical tree cover

Corresponding Author: Dr Douglas Kelley

Version 0:

Decision Letter:

Dear Dr Kelley,

Please allow me to sincerely apologise for the long delay in sending a decision on your manuscript titled "Observational constraints of fire, environmental and anthropogenic on pantropical tree cover". It has now been seen by 3 reviewers, and we include their comments at the end of this message. They find your work of interest, but some important points are raised. We are interested in the possibility of publishing your study in Communications Earth & Environment, but would like to consider your responses to these concerns and assess a revised manuscript before we make a final decision on publication.

We therefore invite you to revise and resubmit your manuscript, along with a point-by-point response that takes into account the points raised. Please highlight all changes in the manuscript text file.

In particular, please ensure your revised manuscript meets the following editorial thresholds:

- * Present a compelling assessment of the influence of limiting factors on global tree cover and clarify how this relates to and advances upon previous understanding of factors influencing tree cover.
- * Fully justify the logic of your approach, including the incorporation of human-modified areas.
- * Clearly explain and describe your statistical methods and ensure that the support for your conclusions is included within the main text and figures, rather than the supplementary information.

Please note that while the Methods section must remain at the end in our format, you can add a short summary of the approach at the end of the Introduction or the start of the Results sections. We also allow up to 10 display items (figures + tables).

Please use the following link to submit your revised manuscript, point-by-point response to the referees' comments (which should be in a separate document to any cover letter), a tracked-changes version of the manuscript (as a PDF file) and the completed checklist:

Link Redacted

We hope to receive your revised paper within six weeks; please let us know if you aren't able to submit it within this time so that we can discuss how best to proceed. If we don't hear from you, and the revision process takes significantly longer, we may close your file. In this event, we will still be happy to reconsider your paper at a later date, as long as nothing similar has been accepted for publication at Communications Earth & Environment or published elsewhere in the meantime.

Please do not hesitate to contact us if you have any questions or would like to discuss these revisions further. We look

forward to seeing the revised manuscript and thank you for the opportunity to review your work.

Best regards,

Joe Aslin

Deputy Editor,
Communications Earth & Environment
<https://www.nature.com/commsenv/>
Twitter: @CommsEarth

EDITORIAL POLICIES AND FORMATTING

Editorial Policy: [Policy requirements](https://www.nature.com/documents/nr-editorial-policy-checklist.pdf) (Download the link to your computer as a PDF.)

Furthermore, please align your manuscript with our format requirements, which are summarized on the following checklist: [Communications Earth & Environment formatting checklist](https://www.nature.com/documents/commsj-phys-style-formatting-checklist-article.pdf)

and also in our style and formatting guide [Communications Earth & Environment formatting guide](https://www.nature.com/documents/commsj-phys-style-formatting-guide-accept.pdf).

***** DATA:** Communications Earth & Environment endorses the principles of the Enabling FAIR data project (<http://www.copdess.org/enabling-fair-data-project/>). We ask authors to make the data that support their conclusions available in permanent, publically accessible data repositories. (Please contact the editor if you are unable to make your data available).

All Communications Earth & Environment manuscripts must include a section titled "Data Availability" at the end of the Methods section or main text (if no Methods). More information on this policy, is available at <http://www.nature.com/authors/policies/data/data-availability-statements-data-citations.pdf>.

If a community resource is unavailable, data can be submitted to generalist repositories such as [figshare](https://figshare.com/) or [Dryad Digital Repository](http://datadryad.org/). Please provide a unique identifier for the data (for example a DOI or a permanent URL) in the data availability statement, if possible. If the repository does not provide identifiers, we encourage authors to supply the search terms that will return the data. For data that have been obtained from publically available sources, please provide a URL and the specific data product name in the data availability statement. Data with a DOI should be further cited in the methods reference section.

REVIEWER COMMENTS:

Reviewer #1 (Remarks to the Author):

This paper presents a global statistical analysis of the controls on tropical tree cover. This is a subject that has received much attention from ecosystem ecologists in the last 15 years. Most find rainfall, fire, and in some places, herbivores or flooding are the strong drivers of tree cover. Interestingly, the authors of this paper suggest that fire has a relatively weak

effect, only likely to reduce tree cover strongly in tropical forests.

The latter point about fires in forests is supported by existing empirical and modeling studies (see Nepstad and Flores citations below) that show increasing fire is a risk to forest cover.

I am somewhat flummoxed by the fire finding, which the authors chose to highlight. It is well-known that fire consumes above-ground woody vegetation, selecting for plants that shunt biomass belowground in grasslands and savannas (e.g., Sankaran et al. 2005; Smit et al. 2010 and others). I don't find that the authors have clearly explained why their model differs from others that show fire to be a strong control on tree cover.

There is possibly some nuance in the results regarding the use of the word "maintaining," though. Is it the authors' contention that maintenance of tree cover in its current state is different from "determining" what that state is? I'm not sure, but that could be clarified, if that is the intention.

I am intrigued by the findings about heat stress and wind throw. These ecological factors are rarely included in large-scale studies of biome distributions or global vegetation structure. Unfortunately, the paper does not make clear to me just how widespread these factors are in space or time. Is there any empirical data that could be marshalled to help? Finally, see my comments below about the inclusion of cropland and urban area. I worry that these are fundamentally different than anything about determinants of tree cover in natural ecosystems and perhaps these areas should be excluded.

Line-by-line comments:

5. I find the title confusing. I don't know what an "observational constraint" is.

60-63. What is "observational constraints?" Does this mean "empirical data?" Does "modelled fire impacts on vegetation are particularly poorly constrained observationally" mean that better field data on the impacts of fire on woody vegetation are needed to parameterize models? A quick search suggests that this term "observational constraints" may be common in physics, but I have never seen it in ecology.

75-77. Not clear why anthropogenically modified areas are included. Row crops, pasture, and urban cover are fundamentally different from natural areas. I would not expect the determinants of tree cover in these areas to be the same as in natural ones. Does the model structure allow for conditional effects akin to interactions in a linear model? That was not clear to me, if so.

79. What does "preservation" mean in this context?

91-94. I think the two measurements described are just absolute and relative percentage change in tree cover, yes?

111. I think it would be helpful to describe in the main text what metric is used for rainfall seasonality. I see several in the methods section.

126-130. It's not new that fire impacts on tree cover are strong in African savannas (e.g., Van Langevelde et al. 2003 and many others). I also find it odd to compare the impacts of a consistent, recurring ecosystem process—fire—with stochastic ones like wind throw.

147-150. Check wording of this sentence.

150-154. Could the inclusion of human-modified areas in the main model be the main reason for the difference relative to studies that find fire to be a stronger influence. People suppress fire in the places they live and farm.

161-165. I am not surprised that fire has a stronger impact on tree cover in areas that have infrequent fire. There is a large literature on the impact of infrequent fires on tropical forest tree cover (e.g., Nepstad et al. 1999; Flores et al. 2016). The wind-throw and heat stress results are perhaps more novel, though the authors have not really highlighted them.

179-184. More explanation is needed in the main text for readers to understand what it means for forests to be more or less sensitive to changes in land use. All forests are sensitive to development—they get cut down.

196. Related to the previous comment, it is very strange to say that land use is a factor "maintaining" tree cover. The only way that would seem to be possible is if there is no human land use, or if there are large areas of forestry, I suppose. Confusing.

325 (Fig. 1). Inclusion of the four human stresses makes little sense to me. It is odd to analyze whether having more pasture leads to having fewer trees. Yes, some pastures (e.g., in the southeastern US) have pockets of tree cover, but generally where there's pasture, it caused removal of nearly all the tree cover that was naturally present. So, is the "percentage reduction" not just the percent of predicted natural tree cover? Same goes for crops, urban area, and human population density, I think.

340 (Fig. 3). Check the caption. It says the maps represent the decrease in tree cover with removal of the left-listed factors. For the top map, that doesn't make sense for human impacts. Removing human impacts would increase tree cover in the Congo Basin and Amazon, where the maps show decreases. Maybe just reverse the caption?

413. Nice that results were reproduced with 5(!) fire datasets.

585. Cite the published version, not the preprint DOI and URL.

Reviewer #2 (Remarks to the Author):

Overall, I found the manuscript a well-designed attempt to partition the impact of fire, heat, humans, and precipitation on tropical tree cover changes globally using remote sensing datasets. I found their approach robust, however, my main critique revolves around their expectation of fire as a source of overstory mortality in many of the ecosystems investigated. For example, in frequently burned woodlands and savannas, overstory mortality is generally minimal, and the frequent surface fire drives the structure of these ecosystems by influencing recruitment through high seedling/sapling mortality but once a tree makes it to a sufficient size, it is generally quite resistant to fire driven mortality. This would result in fire having little explanatory power in terms of loss of overstory tree cover detectable by satellites. If this regular high frequency fire regime were interrupted and fuels allowed to accumulate, then a subsequent fire could initiate high mortality. In this case the fire history as reflected in temporal patterns in burned area are a larger driver of overstory mortality than a single sample of burned area alone. The authors addressed this in the discussion, but it could use more exposition on the role of fire in maintaining structure savanna ecosystems rather than a disturbance alone. Also, it seems that there would be complex spatial interactions among the variables in the case of fire influencing ecotones or temporally where there are interactions between human ignitions or lack thereof and droughts. Further exposition of these principles is needed. The work was excellent and thorough and their techniques robust.

Reviewer #3 (Remarks to the Author):

This paper introduces an interesting analysis on the relative and absolute contributions of various factors that can explain tree cover on a global scale. The main text is well written and reads convincing. But I found it hard to find the evidence of this convincing text from the presented results. For a large part this was because the methods section is much less strong in its wording than the main text. Besides, the approach to highlight the main results, and have the methods at the end (I know this is the format for this journal) didn't work out well in the current version. As a reader you need to have some idea of what you're looking at to make a fair assessment of all the figures that are presented to support the story. I found it also disturbing that next to that the 'methods after results' set-up was not dealt with in a good way, immediately from the beginning most justification was based on figures in supplementary materials. This really needs to be improved. This means that all the justification that is required to judge the paper is coming from details tucked away in an additional document, or has to be derived from a not so clearly written methods section.

Some issues unclear in the methods section apart from the overall methods description:

- The methods section should also better explain how the spatial maps are derived from an essentially non-spatial statistical approach. It is not well described how the methods are placed in a spatial context.
- It is not clear how independent the tree cover predictions provided in with the new method are independent from MODIS VCF data. In fact it was completely unclear to me on what data the Bayesian based estimates of tree cover were based in the first place.
- Related to the fact that it concerns spatial data, it is unclear how data has been controlled for spatial autocorrelation and whether this would have an impact on the analysis (this depends also on the sampling scheme employed). I can imagine this will influence the estimation of prior distributions. If that is not to be expected I would like to see a justification for that.

So I see a lot of potential for this paper when the methods section can be made much clearer and by that more convincing.

Communications Earth & Environment is committed to improving transparency in authorship. As part of our efforts in this direction, we are now requesting that all authors identified as 'corresponding author' create and link their Open Researcher and Contributor Identifier (ORCID) with their account on the Manuscript Tracking System prior to acceptance. ORCID helps the scientific community achieve unambiguous attribution of all scholarly contributions. You can create and link your ORCID from the home page of the Manuscript Tracking System by clicking on 'Modify my Springer Nature account' and following the instructions in the link below. Please also inform all co-authors that they can add their ORCIDs to their accounts and that they must do so prior to acceptance.
<https://www.springernature.com/gp/researchers/orcid/orcid-for-nature-research>

If you experience problems in linking your ORCID, please contact the [Platform](http://platformsupport.nature.com/)

Support Helpdesk.

Version 1:

Decision Letter:

Dear Dr Kelley,

Your manuscript titled "Observational constraints of fire, environmental and anthropogenic on pantropical tree cover" has now been seen by our reviewers, whose comments appear below. In light of their advice we are delighted to say that we are happy, in principle, to publish a suitably revised version in Communications Earth & Environment.

We therefore invite you to revise your paper one last time to address the remaining concerns of our reviewers. At the same time we ask that you edit your manuscript to comply with our format requirements and to maximise the accessibility and therefore the impact of your work.

EDITORIAL REQUESTS:

******Please take care to match our formatting and policy requirements. We will check revised manuscript and return manuscripts that do not comply. Such requests will lead to delays. ******

SUBMISSION INFORMATION:

OPEN ACCESS:

Communications Earth & Environment is a fully open access journal. Articles are made freely accessible on publication. For further information about article processing charges, open access funding, and advice and support from Nature Research, please visit <https://www.nature.com/commsenv/open-access>

Link Redacted

Best regards,

Joe Aslin

Deputy Editor,
Communications Earth & Environment
<https://www.nature.com/commsenv/>
Twitter: @CommsEarth

REVIEWERS' COMMENTS:

Reviewer #1 (Remarks to the Author):

The authors have done an admirable job of addressing my (not limited) concerns in the first round of review. In particular, their clarification of the spatial scale of analysis (50x50 km) helps me to understand a lot of their conclusions in a better light. Several terminology issues have been cleaned up, too.

In general, I still find the design of many figures and tables make them quite hard to understand. For example:

Fig. 1: Caption references left and right columns, but I see six maps top to bottom. If the columns referenced are in the 3x3 square legends, that's not clear. What are the little and big "t" and "T" in the legend with little and big dots next to them?

Figs. 2 & 3 are very hard to and table 1 seems to be missing explanation of the colors.

Reviewer #2 (Remarks to the Author):

The authors should be commended on an excellent job of addressing the reviewer critiques in a thoughtful and thorough manner. This is a very timely and novel approach to a complex question that will generate considerable discussion and it was a pleasure to review.

Reviewer #3 (Remarks to the Author):

The paper has considerably improved since the last time, when it comes to clarity of the methods followed. However, I still think it needs a number of improvements before it can go to a prestigious journal like Nature communications. I will highlight these point by point below. But the main point is that the paper presents (as also explicitly mentioned in the abstract) new insights into what limits tree cover, based on a novel approach. So this new approach is a key element in this paper. I understand that the format of the paper is results first methods later, and I commented on that in the previous round as well, but I still feel the way the methods are now presented (and the way some of the results are explained in the figure captions) does not do justice to this fact. Actually, a main reason why I started to understand the methods is by delving into the earlier paper by the same authors (ref 27), where it is much better explained, and I wonder why that was not done in a similar way here?

So in my opinion, the paper needs a final brush up before it can be published.

Specific comments:

Lines 104-111: The use of terminology is confusing in this section. It is not clear what the difference is between "using data as reference for fitting" versus "incorporating data as input". Likewise, the meaning of "optimization against observed data" is unclear. Does "observed data" then refer to MODIS VCF? I guess so. I understand (after quite some studying of also the referred article 27) that you probably fit (or tune?) the model on the basis of the "distribution of TC as estimated with MODIS VCF", but to me that is still "incorporating data into your models". So you used MODIS VCF for both estimating the model parameters and assessing how well they then reproduce the data. Fair enough, probably there is no alternative when you want to do an analysis at the extent you present here but then this should be made explicit and acknowledged. So stating at the end of this section that "predictions [...] are based not on MODIS VCF data but on the model's optimization against observed tree cover data." Where I assume "observed tree cover data" are the same thing as MODIS VCF, gives a false impression of independence between validation and training data.

Line 113: In line with the above comment, the notion of "test observations" is not clearly defined. Also, I found supplementary figure 2 very hard to assess. I understand now that on the x-axis are NME values, and y-axis is frequency (Basically these are histograms?) but what step 1-3 refers to, and how exactly $\text{sim}(\beta)$ links to $P(X|\beta)$ remains unclear from the figure caption or the above description.

Line 165: The Caption from supplementary Table 1 is confusing. The legend/caption of this table confuses me. If I read this, it for example means that the probability that the impact of burnt area (in the row) exceeds the impact of MAP (in the column) = 1. This means Fire has a BIGGER impact, not a smaller impact as suggested in the text. Or the explanation for this table is not correct (for example, these values represent P-values, and hence < 0.05 is the norm?)

Page 22 Figure 3, similar issue as with previous case. Perhaps provide an example to make the figure understandable. So for the first map, is it correct that the impact of heat stress is almost 100% more likely than the impact of burnt area? Or is it the reverse?

Line 449 Figure 4, Make the titles in the figure better. Now the top one on the left reads just "tree cover" (probably evergreen?) and the one on the right reads "Deciduous cover", but without "fire impact on.." in it. Try to make this consistent with the description below for ease of interpretation.

Line 529: This equation does not represent a logistic curve at all, and also does not match the way it is represented in the references article (ref 27)

Line 535: when you perform a log transformation, approaching 0 goes to negative infinity, it is not clear to me how that corresponds to making equation 1 go to 0 unless the assumption stated implies that there are no values of 0, but the lowest value for MAP is 1. If you mean including an infinitely negative exponent in the logistic equation (which is not what we currently see in Eq1), then indeed you would go to 0. But then this needs to be corrected.

Line 543: It reads "...for $f=SW, LU, p[i]=1$ ". I Assume, that you mean $p[i] \neq 1$ (or perhaps $p[i] > 1$) for S?

Line 553: The caption of supplementary figure 6 is not grammatically correct

Line 555: there is no supplementary figure 12. I assume you mean figure 8

Line 577: What kind of resampling was done? Average? Maximum? Nearest Neighbour? Average may seem a logical choice, but I could also imagine that using the maximum is quite a good way, as this may have a much bigger impact on tree cover than the average. So this also needs a justification

Line 626-626: there seems to go something wrong with the equation formatting (perhaps mixup between word-equations and lateX-type equation formatting), but this needs to be cleaned up. This accounts for multiple equations. For example equation 9 has too few right-sided brackets etc.

We would like to thank the editor and reviewers for their valuable feedback on our study. We appreciate the constructive comments and positive reviews. The majority of the feedback was focused on the methods and methodological choices we made. We have taken this feedback on board and have extensively revised the manuscript to provide a clearer summary of the methods we used and why we chose them. We have also addressed the scales tested and inference methods used, as well as the inclusion of human landscape modification and potentially co-varying predictors.

Thank you again for your help and guidance.

Our responses are in blue, and quoted text from the ms is either in quotation marks or, for longer extracts of text, indented below our responses.

Reviewer #1 (Remarks to the Author):

This paper presents a global statistical analysis of the controls on tropical tree cover. This is a subject that has received much attention from ecosystem ecologists in the last 15 years. Most find rainfall, fire, and in some places, herbivores or flooding are the strong drivers of tree cover. Interestingly, the authors of this paper suggest that fire has a relatively weak effect, only likely to reduce tree cover strongly in tropical forests. The latter point about fires in forests is supported by existing empirical and modeling studies (see Nepstad and Flores citations below) that show increasing fire is a risk to forest cover.

Thank you for the suggested references. We have now added them to the m/s.

I am somewhat flummoxed by the fire finding, which the authors chose to highlight. It is well-known that fire consumes above-ground woody vegetation, selecting for plants that shunt biomass belowground in grasslands and savannas (e.g., Sankaran et al. 2005; Smit et al. 2010 and others). I don't find that the authors have clearly explained why their model differs from others that show fire to be a strong control on tree cover.

Thank you for highlighting these studies, and we agree that it is important to establish why we reached different conclusions.

Sankaran et al. (2005) ¹ discuss the tree cover variation in savannas, identify the limiting factors, and explore the role of disturbances, including fire and herbivory. The main limiting factor Sankaran finds for tree cover potential is maximum precipitation. Fire is the secondary control for MAP >365 mm. However, like others cited in the introduction, Sankaran did not consider other co-varying disturbances or tree cover controls, such as rainfall seasonality or temperature stress. We now add this point to our discussion:

We were able to separate out the effects of different co-varying impacts – which is challenging in many field-based and empirical studies that often consider fire in isolation from other dry disturbances ¹.

When examining studies quantifying fire impacts on tree cover, the most robust evidence often stems from experimental fires, such as the one conducted by Smit (2010) ² in Kruger National Park, South Africa, which assessed the effect of fire on tree cover within a rainfall range of 350-750 mm. However, this study again did not consider co-varying stresses.

Despite these causal limitations of the two studies, the impact of fire on tree cover presented by both studies is not very different from our own. Although direct comparisons with Smit et al. (2010) are challenging due to differences in scale and the binary nature of their experiment, Smit (2010) similarly documented a reduction in tree cover ranging from 1-20% depending on fire treatment, aligning closely with our findings. Additionally, Sankaran et al. proposed a tree cover reduction of approximately 0-20% for fire return intervals spanning 1-10.5 years. Our study demonstrates a tree cover reduction of 0-20% at around 20% annual burnt area, roughly equivalent to a return interval of 5 years, falling well within the range suggested by Sankaran et al. We attribute the reported low tropics-wide impact on tree cover to the fact that regions with annual burnt areas exceeding 20% are confined to relatively small areas with minimal tree cover in the absence of fire, and add to the discussion:

The slight reduction in tree cover could be because of coarse (0.5° x 0.5°) spatial scale. **Other studies show a substantial impact (up to 20%,¹) of fire on tree cover at fire return intervals of around 1 – 10 years, which would result in coarse-scale burnt areas of (1/return time) 10-100% burnt area. Here, we show tree cover is reduced only slightly less than the annual average area burnt - consistent with these finer-scale studies and on par with the impact of agricultural land use per unit area (Supplementary Fig. 9).**

Veenendaal et al 2018 (³ cited in our paper) used this and other fire experiments to quantify the effect of fire return time and seasonality on woody cover. Veenendaal et al. also demonstrate in this study the importance of land conversion preceding the fire regime as a highly important factor. We would argue that the findings in our study are thus in line with the findings in Veenendaal et al' s analysis.

There is possibly some nuance in the results regarding the use of the word “maintaining,” though. Is it the authors’ contention that maintenance of tree cover in it’s current state is different from “determining” what that state is? I’m not sure, but that could be clarified, if that is the intention.

When we talk about 'maintaining' tree cover, we mean that the current geographical conditions are keeping the forest cover at their current levels. This is different from 'determining' tree cover, which involves both past (e.g., fires that remove trees) and current factors (e.g., land-use that replaces trees after those fires) in describing the present tree cover. We have clarified what we mean by both maintaining and determining tree cover in the introduction, so readers are not confused when reading through the results:

For the purpose of this analysis, we consider "maintenance of tree cover" as the ongoing factors that support or suppress the existing tree cover. This means understanding the factors that uphold the current state of tree cover, rather than focusing on the dynamic changes in tree cover over time. This is different from “determination” of tree cover which can result from both previous and present conditions. For example, how fires can be employed as a method of land clearance and transitioning forests to land use ^{4,5}.

I am intrigued by the findings about heat stress and wind throw. These ecological factors are rarely included in large-scale studies of biome distributions or global vegetation structure. Unfortunately, the paper does not make clear to me just how widespread these factors are in space or time. Is there any empirical data that could be marshalled to help?

Heat induced tree mortality events due to combined heat and drought stress have been documented globally ^{6,7}. But, although there is research indicating that hotter temperatures alone can induce stress independent of precipitation amount ⁸ some species show significant tolerance to heat stress and, in most circumstances, when water is unlimited, trees can cope ($T > 40$ °C) for short periods of time .

Windthrow due to extreme winds is a natural disturbance occurrence causing mortality at the scale of individual wind-thrown trees to large blowdown patches ⁹. The impact increases in fragmented forests ¹⁰.

We have incorporated these references in the discussion section: “Heat stress and windthrow have a substantial impact on tree cover **which is in line with** ^{6,7} .”

There are also a handful of studies in tropical forests showing that biotic attacks, following heat-drought, increase mortality ⁹. However, large-scale mortality as observed in temperate and boreal forest have not been documented in the tropics. We have now highlighted that models do not consider this process, stating in the discussion “Also biotic attacks following drought and/or windthrow events ⁹ are not represented in models.”

Finally, see my comments below about the inclusion of cropland and urban area. I worry that these are fundamentally different than anything about determinants of tree cover in natural ecosystems and perhaps these areas should be excluded.

See our comments below on why their inclusion is justified, even if their only effect is to exclude natural vegetation (which many studies show they do not).

Line-by-line comments:

5. I find the title confusing. I don't know what an “observational constraint” is.

“Observational constraint” is a common term used in modelling. It is where we constrain the possible relationship between response (i.e. tree cover) to conditions (i.e. burnt area) through statistical methods, direct observations or measurements in real-world conditions. That fire should impact tropical tree cover by 0.2-3.2% is the constraint that modellers can use to evaluate or optimise their models. However, we realise that this term may not be common across disciplines, so changed the title to “Fire, environmental and anthropogenic controls on pantropical tree cover”

60-63. What is “observational constraints?” Does this mean “empirical data?” Does “modelled fire impacts on vegetation are particularly poorly constrained observationally” mean that better field data on the impacts of fire on woody vegetation are needed to parameterize models? A quick search suggests that this term “observational constraints” may be common in physics, but I have never seen it in ecology.

See above. We've added an "until now" to "modelled fire impacts on vegetation are **until now** particularly poorly constrained observationally" to make it clear that there is a lack of previous attempts at obtaining relationships between stresses and observed tree cover in a way useful for global models. We have added some words to better define observational constraints. Like in physics, observational constraint in global change modelling is any approach that can provide a target range for testing or optimizing an aspect of a model – in our case burnt area's impact on tree cover. This can be empirical data, but on the scales of global vegetation models, statistical approaches such as this for separating the signal from remote sensed observations are also useful. Infact, due to the mismatch in scales, it is often more appropriate to use statistical methods to extract relationships from remote senses observations than rely on "better field data" when constraining global vegetation models. To avoid confusion, though, we have removed the phrase "observational constraint" and now the sentence reads:

However, there is a lack of empirical or observational data or studies that can directly inform the relative importance of these controls on FDGVM resolutions across the tropics.

75-77. Not clear why anthropogenically modified areas are included. Row crops, pasture, and urban cover are fundamentally different from natural areas. I would not expect the determinants of tree cover in these areas to be the same as in natural ones. Does the model structure allow for conditional effects akin to interactions in a linear model? That was not clear to me, if so.

The framework makes similar assumptions to DGVMs to provide a model-relevant constraint. Most DGVMs model trees in the same way in natural and anthropogenically modified landscapes (note that almost none of the world's forests can still be described as entirely "natural"). This assumption is also applied to purely "natural" tree cover, which may behave very differently because of evolutionary history, bioclimatic factors, soils, etc. Our framework accounts for these different interactions in it's Bayesian implementation – if these assumptions have a large effect, we would see larger uncertainty bounds in our results. As we do have large uncertainty ranges, an interesting extension of this study could be to test if differing tree interactions along the anthropogenically modified gradient in the future highlighted by the reviewer may help constrain our results further. And we have added this to the discussion:

Like DGVMs, our model considers similar responses of tree cover across continents with different evolutionary histories and across gradients of anthropogenic landscape modification. Further framework development could attribute uncertainties in tree cover disturbance responses to these assumptions, which may also help guide DGVM development priorities.

However, despite the large uncertainties, our results are constrained enough to show some surprising results for the fire that are important for global change and the DGVM community.

79. What does "preservation" mean in this context?

This is a typo, and we have re-written the sentence to better describe the advantage of using this technique:

The technique's **comprehensive** quantification of uncertainty helps explore **highly** correlated factors influencing tree cover such as tropical fires with seasonal drought-induced vegetation stress and co-varying land use (LU) and fire.

91-94. I think the two measurements described are just absolute and relative percentage change in tree cover, yes?

Correct, and we have added the word “absolute” to describe the first:

When analysing the impact of different factors on tree cover, we need to consider two key measurements. The first is the **absolute** difference (or impact) on tree cover with and without the influence of the factor in question. The second is the **relative** impact on tree cover, measured as the difference in tree cover as a percentage of the original tree cover before the factor was introduced.

111. I think it would be helpful to describe in the main text what metric is used for rainfall seasonality. I see several in the methods section.

We have added the following two sentences:

We use four metrics as a proxy for rainfall seasonality: Fractional mean annual dry days, fractional number of dry days in the driest month, precipitation in the driest month, and mean seasonal precipitation concentration.

126-130. It’s not new that fire impacts on tree cover are strong in African savannas (e.g, Van Langevelde et al. 2003 and many others). I also find it odd to compare the impacts of a consistent, recurring ecosystem process—fire—with stochastic ones like wind throw.

While fire may exhibit consistent recurrence in certain ecosystems, it can manifest as a stochastic process in others. However, our analysis aims to detect emergent signals of different drivers, including fire and wind throw, on tree cover over coarse scales and extended periods of time. It is important to note that even stochastic processes, when observed on such scales, tend towards exhibiting recurrence patterns, albeit not necessarily cyclical. Moreover, our choice of Bayesian methods is deliberate and well-suited for handling both stochastic and predictable drivers.

We have added the following towards the end of the introduction to make this point clear:

The Bayesian approach produces a generative modelling framework that inherently accounts for stochasticity ¹¹, enabling the exploration of unpredictable factors', such as wind throw and in certain ecosystems, long-return time fires, and how they impact on tree cover.

147-150. Check wording of this sentence.

We have modified the sentence to read:

We quantify fire impact on tree cover without human influence by simulating the impact on tree cover from the burnt area we would expect to observe without human modification, which we obtained from ¹² (see methods) and without population density or land use influence on tree cover (Fig. 5).

150-154. Could the inclusion of human-modified areas in the main model be the main reason for the difference relative to studies that find fire to be a stronger influence. People suppress fire in the places they live and farm.

FireMIP models also include human landscape modification and influence on burnt area^{13,14}. If they simulate both processes correctly, then fire's influence on tree cover should be close or within the original constraint range of 0.2-3.2%. Our study additionally analyses the impact fire has without human influence on either the landscape or burnt area, which would account for both the suppressive effects of fire (which dominate in the Kelley et al. study we obtained without human burnt area from) as the reviewer suggests. This does show that fire could have more of an impact without humans, and we have modified the discussion to reflect this:

Fire has a surprisingly low influence on tropics-wide tree cover, though it plays a more important role in savannas, suppressing tree cover by 0.6-7.1%. **Without human impact on burnt area and tree cover loss, the impact on fire in savannas has the potential to be much higher at between 0.8-10.8% (Table 1). This is more in line with empirical studies and field experiments^{1,2}, though it still shows** that the independent impact of fire is not enough to fully explain the lack of tree cover within savannas.

161-165. I am not surprised that fire has a stronger impact on tree cover in areas that have infrequent fire. There is a large literature on the impact of infrequent fires on tropical forest tree cover (e.g., Nepstad et al. 1999; Flores et al. 2016).

We agree that, ecologically, this is not surprising, and we have added the suggested references to clarify this:

However, as these areas have not historically experienced high fire occurrence, their vegetation is unlikely to be fire-adapted. And many studies have highlighted tropical forest areas that experience infrequent burning as particularly sensitive to even small increases in burning¹⁵⁻¹⁷.

We emphasise on this point in our study to aid prioritisation for DGVM development – i.e DGVMs should focus on the impact of fire on tree cover in fire-vulnerable ecosystems rather than using fire as a maintenance variable in savanna areas. This is clearly stated in the closing sentence of the conclusions:

Therefore, based on our results, we recommend targeting FDGVM development to simulate fire and vegetation responses to burning in tropical forested areas - essential for evaluating future impacts of fire in these carbon-rich forests.

The wind-throw and heat stress results are perhaps more novel, though the authors have not really highlighted them.

See the response to general comments for studies that have discussed some of these effects. What is surprising is that they have more of an impact on tree cover than fire, which is why we have chosen to present these results in relation to burnt area. However, the reviewer is correct that we also offer constraints that may be used by models simulating these other processes. We have strengthened this point in the discussion:

The constraints we have found on the impact of wind and heat stress could be particularly useful to assess and re-parameterise vegetation models that represent both these disturbances.

179-184. More explanation is needed in the main text for readers to understand what it means for forests to be more or less sensitive to changes in land use. All forests are sensitive to development—they get cut down.

When we are looking at the sensitivity, we aren't just interested in the direction of change as the reviewer implies, but the rate of change as our controls increase or decrease. There are many effects land use change can have on tree cover, which affects this rate of change. These could include:

1. **Spatial Heterogeneity:** Land use changes may not occur uniformly across a landscape. For example, while there might be a 10% increase in agricultural land overall, this change may not be evenly distributed. Some areas may experience intensive conversion to agriculture, while others remain unchanged or undergo different land use changes. Some land use conversions may preferentially target non-forested areas and have less of an impact. As a result, the impact on tree cover may vary spatially, leading to disparities in land use change and resultant observed changes in tree cover.
2. **Differential Sensitivity to Land Use Changes:** Different types of vegetation may respond differently to changes in land use. For instance, mature forests may be less susceptible to conversion than secondary forests or shrublands. Therefore, even if there is a significant increase in agricultural land, the impact on tree cover may be moderated by the resilience of different vegetation types to land use changes.
3. **Ecological Resilience:** Degraded lands may experience further degradation due to factors such as soil erosion or invasive species invasion, exacerbating the loss of tree cover beyond the direct impact of land use changes.
4. **Fragmentation Effects:** Land use changes, particularly those involving habitat fragmentation, can influence tree cover dynamics beyond the extent of land conversion. Fragmentation disrupts ecological connectivity and alters microclimatic conditions, leading to changes in species composition, habitat quality, and ecosystem functioning. In fragmented landscapes, edge effects, such as altered moisture levels, can impact tree growth and survival near the forest edge. As a result, even if the overall extent of land conversion is relatively small, the spatial configuration and fragmentation of remaining forest patches can exacerbate the loss of tree cover, contributing to disparities between observed changes in land use and tree cover.
5. **Management Practices:** Land management practices, such as agroforestry or reforestation initiatives, can influence tree cover independently of changes in land use. For instance, afforestation efforts in agricultural landscapes or reforestation programs in degraded areas may lead to increases in tree cover despite ongoing agricultural expansion.

This will not be an exhaustive list, but demonstrates that “they get cut down” does not capture the full range of impact land use has on tree cover.

196. Related to the previous comment, it is very strange to say that land use is a factor “maintaining” tree cover. The only way that would seem to be possible is if there is no human land use, or if there are large areas of forestry, I suppose. Confusing.

There might also be some confusion of scale here. If we were to look at a specific point, then yes, land use would maintain very little or no tree cover (though noting that pasture areas can still include woody cover). However, as clearly stated in the introduction and methods, we are looking at coarse scales in this study. Therefore, land use is represented as a percentage of cover, and the impact of land use on tree cover—both locally and from nearby land use, as discussed in the previous point—can safely be assumed to happen within the same gridcell.

We have made this concept more explicit throughout. We have:

1. Explicitly stated the resolution of our framework in the introduction (“we test the climate and environmental influences on tropical tree cover using a Bayesian limitation framework optimised against tree cover observations **on a typical DGVM resolution of 0.5 (roughly 50 by 50km at the equator).**”)
2. Add “percentage” to the sentence “(4) human pressure expressed as land use (**percentage** cropland, pasture, urban cover) and population density” to make it clear that our land use is not a discrete variable.
3. Included the following paragraph in the discussion:

The impact of land use on tree cover does not always match the extent of the tree cover itself. This is because, in addition to the extent of land use changes, reductions in tree cover may diverge due to various factors, including spatial heterogeneity, differential sensitivity of vegetation types, ecological resilience and regeneration processes, fragmentation effects, management practices, and climate and environmental factors. While some of these impacts extended beyond the land cover extent itself, we safely assumed that they occur within the same gridcell, given the coarse scale of analysis (0.5°, ~ 50km) employed in this study. That tree cover responses follow cropland extent (Fig. 6), suggesting that any additional impact on tree cover beyond cropland extent are negligible on our coarse scales. Urban areas do have a large impact beyond their extent – up to 10 times are lower urban covers, possibly owing to factors such as heat island effects, altered microclimates, fragmentation of surrounding ecosystems, and direct human disturbances such as deforestation and land clearing for urban expansion. Pasture, however, shows a smaller impact than pastures own extent, especially as pasture area increases, indicating high tree cover retention at higher pasture areas.

325 (Fig. 1). Inclusion of the four human stresses makes little sense to me. It is odd to analyze whether having more pasture leads to having fewer trees. Yes, some pastures (e.g., in the southeastern US) have pockets of tree cover, but generally where there’s pasture, it caused removal of nearly all the tree cover that was naturally present. So, is the “percentage reduction” not just the percent of predicted natural tree cover? Same goes for crops, urban area, and human population density, I think.

Our framework can represent linear reduction in tree cover from specific drivers and serves as a test of this hypothesis. Even if the only effect of human stresses is to remove tree cover that is naturally present, then the current approach is still justified. In Figure S9 (now main Figure 7), if the responses stay at 1, then this implies that this variable only serves to remove natural tree cover within its area. If the response is above one, such as for an urban area, then this implies the removal of tree cover above that area. For urban, this makes sense and is well documented. Increased urbanisation brings Urban Heat Island Effects, fragmentation of ecosystems and distribution of critical ecological

processes making them more vulnerable to other stresses, increased pollution, disrupt natural hydrological cycles, demand for building materials and mining, all leading to a decline in tree cover beyond urban areas ^{4,18–22}.

For pasture, the response is above 1 for low pasture areas and below one for greater, meaning the marginal impact of pasture areas decreases with land conversion. This also makes sense. In areas with low levels of pasture, the ecosystem may be more intact and less disturbed. Therefore, the introduction of even small amounts of pasture and fragmentation can have a significant impact on the environment, disrupting native vegetation, altering soil properties, and affecting wildlife habitats ^{4,22}. As pasture levels increase, the cumulative impact on the environment also increases, but the rate of increase may slow down over time. This is because many of the most sensitive or easily affected components of the ecosystem may have already been impacted at lower pasture levels, leading to diminishing returns in terms of additional impacts as pasture levels rise.

340 (Fig. 3). Check the caption. It says the maps represent the decrease in tree cover with removal of the left-listed factors. For the top map, that doesn't make sense for human impacts. Removing human impacts would increase tree cover in the Congo Basin and Amazon, where the maps show decreases. Maybe just reverse the caption?

The caption is correct but not as clear as it could be. We have rewritten as follows:

Fig. 5. The percentage reduction in tree cover area by fire. (top) Fire as per Fig. 2, (2nd row) fire on tree cover without the impact of rainfall distribution (3rd row) fire without direct human influence on tree cover and (bottom) fire without influence from human impact on tree cover or burnt area. Columns show 10% and 90% percentiles accounting for framework uncertainty.

413. Nice that results were reproduced with 5(!) fire datasets.

Thank you very much

585. Cite the published version, not the preprint DOI and URL.

Corrected

Reviewer #2 (Remarks to the Author):

Overall, I found the manuscript a well-designed attempt to partition the impact of fire, heat, humans, and precipitation on tropical tree cover changes globally using remote sensing datasets. I found their approach robust, however, my main critique revolves around their expectation of fire as a source of overstory mortality in many of the ecosystems investigated. For example, in frequently burned woodlands and savannas, overstory mortality is generally minimal, and the frequent surface fire drives the structure of these ecosystems by influencing recruitment through high seedling/sapling mortality but once a tree makes it to a sufficient size, it is generally quite resistant to fire driven mortality. This would result in fire having little explanatory power in terms of loss of overstory tree cover detectable by satellites. If this regular high frequency fire regime were interrupted and fuels

allowed to accumulate, then a subsequent fire could initiate high mortality. In this case the fire history as reflected in temporal patterns in burned area are a larger driver of overstory mortality than a single sample of burned area alone. The authors addressed this in the discussion, but it could use more exposition on the role of fire in maintaining structure savanna ecosystems rather than a disturbance alone.

We thank the reviewer for suggesting one of the mechanisms by which burnt area would have a small impact. We use the expectation /assumption used in most fire-enabled DGVMs ^{13,14,23,24}. While most (though not all) FDGVMS would technically consider the fuel dynamics described for the reviewer, the effects on tree cover can be very well explained in several fire models using just burnt area as an explanatory mortality factor ^{14,24,25}.

We have introduced this concept in the discussion and noted this as a possible explanation for the small impact of fire on tree cover in the tropics:

The use of remotely sensed data may contribute to the surprisingly low impact. Overstory mortality is generally minimal in frequently burned woodlands and savannas, with frequent surface fires primarily influencing recruitment through high seedling/sapling mortality. Therefore, the observed low impact of burnt areas from surface fires on tree cover, as detected by satellites, may reflect the resilience of mature trees to fire-driven mortality in these environments. However, it is worth noting that FDGVMS tend to target remote sensed burnt area for parameterisation and evaluation ^{13,14,23}.

Also, it seems that there would be complex spatial interactions among the variables in the case of fire influencing ecotones or temporally where there are interactions between human ignitions or lack thereof and droughts. Further exposition of these principles is needed.

This is the reason for using Bayesian inference to optimise our model. Given the scales we wish to inform and the available information, assessing these interactions from our data would be hard. Bayesian techniques allow us to quantify the uncertainty that these missing processes have on our results ^{11,26}. Not representing these processes probably contributes to our wide uncertainty ranges. However, our results are still robust, even accounting for this uncertainty. We have modified the discussion to reflect this. (See response to reviewer 1's comment on line 196)

The work was excellent and thorough and their techniques robust.

Thank you.

Reviewer #3 (Remarks to the Author):

This paper introduces an interesting analysis on the relative and absolute contributions of various factors that can explain tree cover on a global scale. The main text is well written and reads convincing. But I found it hard to find the evidence of this convincing text from the presented results. For a large part this was because the methods section is much less strong in it's wording than the main text. Besides, the approach to highlight the main results, and have the methods at the end (I know this is the format for this journal) didn't work out well in the current version. As a reader you need to have some idea of what you're looking at to make a fair assessment of all the figures that are

presented to support the story. I found it also disturbing that next to that the 'methods after results' set-up was not dealt with in a good way, immediately from the beginning most justification was based on figures in supplementary materials. This really needs to be improved. This means that all the justification that is required to judge the paper is coming from details tucked away in an additional document, or has to be derived from a not so clearly written methods section.

We have expanded the summary of the methods to cover a full modelling framework description, including how the underlying model was contrasted, information on how the model was optimized, dataset information, and the advantages of using Bayesian techniques for observational constraints assessment. The full methods summary now reads:

To optimize the framework against MODIS Vegetation Continuous Fields (VCF) collection 6 fractional tree cover²⁷, we used a Bayesian inference technique. Bayesian inference allows us to assess the likely range of each control's impact by removing their influence. The technique's comprehensive quantification of uncertainty helps explore highly correlated factors influencing tree cover^{12,28}, such as tropical fires with seasonal drought-induced vegetation stress^{29,30} and co-varying land use (LU) and fire^{25,31,32}. The Bayesian approach produces a generative modelling framework that inherently accounts for stochasticity¹¹, enabling the exploration of unpredictable factors', such as wind throw and in certain ecosystems, long-return time fires, and how they impact tree cover. This optimization was conducted across the tropics, leveraging spatial variations in tree cover to inform our analysis. As MODIS VCF data was used as a reference for model fitting and evaluation but were not directly incorporated into the model as input data, tree cover predictions generated by the framework are based not on MODIS VCF data but on the model's optimization against observed tree cover data.

Our framework produces probability distributions of simulated tree cover. It compares well against test observations (see evaluation supplement and Supplementary Fig. 2, Supplementary Fig. 3). Where observations of tree cover align with the predicted range (Supplementary Fig. 4), the model correctly reproduces the controls' influence on tree cover and its uncertainty, and indicates the model captures the uncertainty of the controls' influence over tree cover²⁸. We report changes in cover at the 10-90% percentile confidence range of the framework's posterior probability distribution, which provided a range of plausible constraints on each factor's impact. Assumptions about noninteracting factors are included in uncertainty ranges¹¹. See methods for full framework description.

When analysing the impact of different factors on tree cover, we need to consider two key measurements. The first is the absolute difference (or impact) on tree cover with and without the influence of the factor in question. The second is the relative impact on tree cover, measured as the difference in tree cover as a percentage of the original tree cover before the factor was introduced.

After assessing how large-scale climate gradients influence tree cover, we use this framework to test how stresses and human impacts limit tree cover tropic-wide and in different vegetation types. We identify fire impact on tree cover as a significantly lower response than found in fire-enabled DGVMs. We demonstrate how our results are consistent with field measurements and fire exclusion experiments before discussing the implications for global vegetation modelling. Finally, we explore where tree cover is sensitive to recent or potential future changes in fire.

We have also added explanations of the methods at relevant parts of the results. i.e:

We use four metrics as a proxy for rainfall seasonality: Fractional mean annual dry days, fractional number of dry days in the driest month, precipitation in the driest month, and mean seasonal precipitation concentration. The results presented here summarize the impact via performance-weighted contributions of all.

And

The impact of land use on tree cover does not always match the extent of the tree cover itself. This is because, in addition to the extent of land use changes, reductions in tree cover may diverge due to various factors, including spatial heterogeneity, differential sensitivity of vegetation types, ecological resilience and regeneration processes, fragmentation effects, management practices, and climate and environmental factors. While some of these impacts extended beyond the land cover extent itself, we safely assumed that they occur within the same gridcell, given the coarse scale of analysis (0.5° , $\sim 50\text{km}$) employed in this study.

We have moved Supplementary Fig. 2, 7, 8, and 9 to the main part of the paper. These figures demonstrate how the modelling framework works and provide valuable information to enhance the paper's narrative.

Some issues unclear in the methods section apart from the overall methods description:

-The methods section should also better explain how the spatial maps are derived from an essentially non-spatial statistical approach. It is not well described how the methods are placed in a spatial context.

While the statistical framework does not incorporate spatial dependencies, it runs perfectly well on spatial data. This is not unusual, the statistical fire model it is based on^{12,26} and many fire land surface schemes, whether statistical, empirical or process-based, run the same way: They receive spatial inputs, perform numerics on these input without spatial dependency, and produce spatial outputs. Specifically, the workflow for our model is:

Input Data: Our input data includes annual average climate variables, human land use, population density, and remotely sensed burnt area maps. Additionally, we use some sub-yearly metrics, such as rainfall distribution. The training data is the MODIS VCF, and all datasets are regridded to a 0.5-degree grid in netCDF format.

Model Application: Our Bayesian model does not incorporate spatial dependence or structure, treating spatial data as a collection of independent grid cells. We train the model parameters using Bayesian inference.

Output Generation: For each experiment, the trained model is rerun with spatial maps of inputs, which are altered depending on the specific experiment. This process involves providing the model with maps as inputs and obtaining spatial maps as outputs. We follow the commonly used technique in Bayesian Inference of approximating the model posterior with 1000 ensemble members, as clearly stated in the methods.

Integration and Visualization: The integration of spatial outputs with statistical results occurs inherently through the model's processing. The resulting spatial maps are directly visualized in the figures presented in the main text.

- It is not clear how independent the tree cover predictions provided in with the new method are independent from MODIS VCF data. In fact it was completely unclear to me on what data the Bayesian based estimates of tree cover were based in the first place.

The study examined the controls influencing changes in tree cover in space. Our approach involves utilizing a set of predictors, termed drivers, which are independent of tree cover observations. These drivers serve as inputs to a model trained on MODIS VCF data of tree cover observations using Bayesian Inference. The model predicts tree cover based on these drivers, allowing us to explore how changes in these drivers affect tree cover dynamics.

Regarding the Bayesian model, it's important to note that the model was fitted against predictions of MODIS VCF, rather than using it as input data directly. This means that the Bayesian model utilizes the predictions from MODIS VCF to optimize its parameters and make predictions of tree cover dynamics.

We have added this to the revised methods summary:

The Bayesian approach produces a generative modelling framework that inherently accounts for stochasticity¹¹, enabling the exploration of unpredictable factors¹, such as wind throw and in certain ecosystems, long-return time fires, and how they impact on tree cover. This optimization was conducted across the tropics, leveraging spatial variations in tree cover to inform our analysis. **As MODIS VCF data was used as a reference for model fitting and evaluation but were not directly incorporated into the model as input data, tree cover predictions generated by the framework are based not on MODIS VCF data but on the model's optimization against observed tree cover data.**

-Related to the fact that it concerns spatial data, it is unclear how data has been controlled for spatial autocorrelation and whether this would have an impact on the analysis (this depends also on the sampling scheme employed). I can imagine this will influence the estimation of prior distributions. If that is not to be expected I would like to see a justification for that.

Our Bayesian inference model does not explicitly account for spatial autocorrelation. This approach aligns with established methodologies in similar studies, such as MaxEnt species distribution models^{33,34}, fire driver Bayesian models^{12,28,35}, and land surface schemes^{24,25,36}, which also do not incorporate spatial dependence. This is because the coarse spatial resolution of 0.5 degrees, covering a large geographic extent from 30 degrees North to 30 degrees South across various ecosystems, ranging from tropics to temperate forests, savannas, grasslands, and deserts, reduces the likelihood of neighbouring grid cells exhibiting similar characteristics based purely on autocorrelation, and therefore we expect its effect to be minimal.

Regarding our sampling scheme, we utilized a Metropolis-Hastings Markov Chain Monte Carlo (MCMC) step with the PyMC3 Python package. We ran 10 chains with 10,000 iterations over 20% of

the data points (N = 2408) for each rainfall seasonality metric and dataset. The random sampling design, involving the random selection of 20% of the data points across the study region, ensures spatial representativeness and reduces the risk of introducing spatial bias into the analysis.

We set broad prior distributions to encompass the full range of physically plausible values, reflecting a lack of prior knowledge and allowing the data to inform the posterior distributions fully. These non-informative priors, coupled with the diverse environmental conditions and limitation-based modelling approach focusing on assessing the relative importance of various controls on tree cover dynamics, may collectively reduce the impact of spatial autocorrelation on the analysis.

So I see a lot of potential for this paper when the methods section can be made much clearer and by that more convincing.

Thank you very much!

1. Sankaran, M. *et al.* Determinants of woody cover in African savannas. *Nature* **438**, 846–849 (2005).
2. Smit, I. P. J. *et al.* Effects of fire on woody vegetation structure in African savanna. *Ecol. Appl.* **20**, 1865–1875 (2010).
3. Veenendaal, E. M. *et al.* On the relationship between fire regime and vegetation structure in the tropics. *New Phytol.* **218**, 153–166 (2018).
4. Lapola, D. M. *et al.* The drivers and impacts of Amazon forest degradation. *Science* **379**, eabp8622 (2023).
5. Ferreira Barbosa, M. L. *et al.* Recent trends in the fire dynamics in Brazilian Legal Amazon: Interaction between the ENSO phenomenon, climate and land use. *Environ. Dev.* **39**, 100648 (2021).
6. Allen, C. D., Breshears, D. D. & McDowell, N. G. On underestimation of global vulnerability to tree mortality and forest die-off from hotter drought in the Anthropocene. *Ecosphere* **6**, 1–55 (2015).
7. Allen, C. D. *et al.* A global overview of drought and heat-induced tree mortality reveals emerging climate change risks for forests. *For. Ecol. Manage.* **259**, 660–684 (2010).

8. Guha, A., Han, J., Cummings, C., McLennan, D. A. & Warren, J. M. Differential ecophysiological responses and resilience to heat wave events in four co-occurring temperate tree species. *Environ. Res. Lett.* **13**, 065008 (2018).
9. McDowell, N. *et al.* Drivers and mechanisms of tree mortality in moist tropical forests. *New Phytol.* **219**, 851–869 (2018).
10. Laurance, W. F. & Curran, T. J. Impacts of wind disturbance on fragmented tropical forests: A review and synthesis. *Austral Ecol.* **33**, 399–408 (2008).
11. Gelman, A. *et al.* *Bayesian Data Analysis*. (Chapman and Hall/CRC, 2013).
12. Kelley, D. I. *et al.* How contemporary bioclimatic and human controls change global fire regimes. *Nat. Clim. Chang.* **9**, 690–696 (2019).
13. Hantson, S., Arneeth, A., Harrison, S. P. & Kelley, D. I. The status and challenge of global fire modelling. (2016).
14. Rabin, S. S., Melton, J. R. & Lasslop, G. The Fire Modeling Intercomparison Project (FireMIP), phase 1: experimental and analytical protocols with detailed model descriptions. *Geoscientific Model* (2017).
15. Nepstad, D. C. *et al.* Large-scale impoverishment of Amazonian forests by logging and fire. *Nature* **398**, 505–508 (1999).
16. Flores, B. M., Fagoaga, R., Nelson, B. W. & Holmgren, M. Repeated fires trap Amazonian blackwater floodplains in an open vegetation state. *J. Appl. Ecol.* **53**, 1597–1603 (2016).
17. Staver, A. C. *et al.* Thinner bark increases sensitivity of wetter Amazonian tropical forests to fire. *Ecol. Lett.* **23**, 99–106 (2020).
18. Browder, J. O. The urban-rural interface: Urbanization and tropical forest cover change. *Urban Ecosyst.* **6**, 21–41 (2002).
19. Miller, M. D. The impacts of Atlanta's urban sprawl on forest cover and fragmentation. *Appl. Geogr.* **34**, 171–179 (2012).

We would like to thank all the reviewers for their insightful comments and constructive feedback. We are pleased to see that the reviewers agree the manuscript is strong overall and that the revisions have provided a much clearer description of the modelling protocol. The remaining detailed critiques have helped us significantly improved the clarity and rigor in the description of our methods and the explanation of our novel framework.

In this round, Reviewer 1 requested specific clarifications and improvements on the figures, tables, and their captions, which we outline under each point. Reviewer 2 did not ask for further changes. Reviewer 3 asked for additional revisions to the description of our method to improve clarity, and so we provide a more detailed response specifically to their review.

Responses are in blue, and text from the manuscript is in quotation marks or, for longer sections, indented text.

Thanks again,

Douglas Kelley on behalf of co-authors.

Reviewer #1 (Remarks to the Author):

The authors have done an admirable job of addressing my (not limited) concerns in the first round of review. In particular, their clarification of the spatial scale of analysis (50x50 km) helps me to understand a lot of their conclusions in a better light. Several terminology issues have been cleaned up, too.

In general, I still find the design of many figures and tables make them quite hard to understand. For example:

Fig. 1: Caption references left and right columns, but I see six maps top to bottom. If the columns referenced are in the 3x3 square legends, that's not clear. What are the little and big "t" and "T" in the legend with little and big dots next to them?

Apologies, this was a formatting issue during m/s upload. The new upload should show the figure in the correct grid and added sub-figure letter labels a)-f) to clarify. We have updated the figure and legend to:

Fig. 1 Limiting controls on tree cover. a-c) shows the relative standard limitation for each control and d-f) normalised sensitivity of each factor. Purple shows areas limited by mean annual environmental stresses (S), yellow by human pressure from population density and land use (L), Cyan by Mean annual Precipitation (P) and dots by Mean Annual Temperature and Shortwave Radiation (T). Red represents co-limitation by S&L, blue by S&P, and green by L&P. Shades show the relative importance of the limitation, with darker, intense shades indicating a stronger impact, lighter shaded (and non-capitalised letter in legend) less impact, and white indicating little or no limitation - by definition coinciding with high tree cover. From top-bottom maximum stress, human pressure, and MAP limitation at 10% likelihood.

Figs. 2 & 3 are very hard to and table 1 seems to be missing explanation of the colors.

We suspect that the reviewer is suggesting Figs. 2 and 3 need better explanations in the caption. We have updated figure captions to read:

Fig. 2 The percentage reduction in tree cover area by each environmental and human stresses. Each row represents a different stress (top-bottom): Fire (using burnt area), heat stress, windthrow, and seasonal rainfall distribution. These are followed by human pressures: population density, urban area, cropland area and pasture area. For each stress or human pressure, two maps are shown: the left map represents the 10th percentile, and the right map represents the 90th percentile of the likely range of the stress' impacts, illustrating the range of uncertainty in the estimated tree cover reduction. This figure allows for a visual comparison of both the magnitude of tree cover reduction by each stress and the confidence level (percentile range) associated with these reductions.

Fig. 3: Pairwise comparison of the likelihood that the stress or human pressure in each column reduces tree cover more than the one in each row. Each map in the grid shows indicates the likelihood of the column stress having a greater impact on tree cover reduction

than the row stress. Blue areas represent regions where the column stress is more likely to cause a higher reduction in tree cover, while brown areas represent regions where the row stress has a higher likelihood. The stress or pressure's first two letters or initials are listed next to the relevant colour for each map. For example, the top left blue areas show where Hs (Heat stress) reduces tree cover more than Ba (Burnt area). White indicates equal likelihood, and lighter shades of blue or brown show a slight likelihood difference between the column and row stress. The color gradients allow for a visual comparison of how different stresses or pressures are likely to impact tree cover in various locations.

And have added “The colours in the table are shaded according to the intensity of impact, highlighting where the stresses or pressures have the greatest effect, with darker shades indicating larger impacts and making the table more accessible” to the end of the caption for Table 1.

Reviewer #2 (Remarks to the Author):

The authors should be commended on an excellent job of addressing the reviewer critiques in a thoughtful and thorough manner. This is a very timely and novel approach to a complex question that will generate considerable discussion and it was a pleasure to review.

Reviewer #3 (Remarks to the Author):

The paper has considerably improved since the last time, when it comes to clarity of the methods followed. However, I still think it needs a number of improvements before it can go to a prestigious journal like Nature communications. I will highlight these point by point below. But the main point is that the paper presents (as also explicitly mentioned in the abstract) new insights into what limits tree cover, based on a novel approach. So this new approach is a key element in this paper. I understand that the format of the paper is results first methods later, and I commented on that in the previous round as well, but I still feel the way the methods are now presented (and the way some of the results are explained in the figure captions) does not do justice to this fact. Actually, a main reason why I started to understand the methods is by delving into the earlier paper by the same authors (ref 27), where it is much better explained, and I wonder why that was not done in a similar way here?

So in my opinion, the paper needs a final brush up before it can be published.

Thank you for your detailed feedback and for highlighting the importance of clearly presenting the novel approach. In response to your comment, we have restructured and expanded key portions of the methods summary section to better align with our earlier work and to clarify the novelty of the Bayesian limitation framework used here. The full revision can be seen in the track changes version of the m/s. Specifically we have:

1. **Expanded the description of the method** by adding more detail on how each limiting factor (precipitation, energy, stress, and human pressure) is modelled and interacts, including describing how we use logistic curves to simulate fractional tree cover, clarifying both individual and combined effects. The explanation now better mirrors the clarity of ref 27 while maintaining the results-first structure.
2. **We have clarified the framework's novelty.** We now emphasise how the Bayesian limitation framework systematically isolates and quantifies the influence of each factor, noting the

advantage of generating probability distributions, accounting for stochasticity, and assessing model uncertainty. This highlights the novelty of our approach, which is distinct from previous modelling and observational studies.

3. **We added further clarity in figure captions.** We revisited the figure captions to ensure they explain how the framework contributes to the study's key results and insights, making the connection between methods and findings clearer. Note that some of these revisions are outlined in response to reviewer 1.

We hope these revisions address your concern and highlight the novelty of our methods, while also enhancing the overall clarity of the paper.

Specific comments:

Lines 104-111: The use of terminology is confusing in this section. It is not clear what the difference is between "using data as reference for fitting" versus "incorporating data as input". Likewise, the meaning of "optimization against observed data" is unclear. Does "observed data" then refer to MODIS VCF? I guess so. I understand (after quite some studying of also the referred article 27) that you probably fit (or tune?) the model on the basis of the "distribution of TC as estimated with MODIS VCF", but to me that is still "incorporating data into your models". So you used MODIS VCF for both estimating the model parameters and assessing how well they then reproduce the data. Fair enough, probably there is no alternative when you want to do an analysis at the extent you present here but then this should be made explicit and acknowledged. So stating at the end of this section that "predictions [...] are based not on MODIS VCF data but on the model's optimization against observed tree cover data." Where I assume "observed tree cover data" are the same thing as MODIS VCF, gives a false impression of independence between validation and training data.

We have made several revisions to the text to address these points:

1. We clarified that the optimization process involved leveraging spatial variability in MODIS VCF data by specifying that 20% of the data was used for parameter optimization and the remaining 80% for validation. This acknowledges the potential overlap between training and validation datasets and addresses concerns about potential bias due to the lack of alternative datasets at this resolution. The relevant part of the text now reads:
"The optimisation was conducted across the pan-tropics between 30° North to South, leveraging the spatial variability of tree cover in MODIS VCF data to inform the model. To evaluate the model's performance, we used 20% of the MODIS VCF data for parameter optimisation and reserved the remaining 80% as validation data. While this overlap between training and validation datasets introduces some potential bias, the lack of alternative global datasets at this resolution makes this approach necessary. Importantly, we focused on understanding the relative impacts of different controls on tree cover, rather than producing absolute predictions. This also mitigates against any potential biases in the MODIS VCF training data."
2. We emphasized that while MODIS VCF data was used to calibrate the model, it was not incorporated directly as input data in tree cover predictions. This distinction helps clarify the relationship between the training and driving datasets. The revised text now includes:

“MODIS VCF data was used to calibrate the model but was not incorporated as input data in tree cover predictions. The Bayesian optimisation generated parameter distributions that inform the final predictions, ensuring that the impacts of climate, stress, and human pressures on tree cover can be quantified independently of the direct use of MODIS VCF data in the final predictions.”

3. Additionally, we elaborated on the Bayesian framework's ability to systematically remove the influence of each control, allowing us to assess how individual factors independently affect tree cover while providing comprehensive uncertainty quantification. We added:
“The modelling approach allows us to systematically remove the influence of each control, enabling the exploration of how each factor independently affects tree cover.” (This line emphasizes how the Bayesian approach accounts for independent factor analysis.)
and
“The Bayesian optimization generates probability distributions for each simulated tree cover scenario, providing comprehensive uncertainty quantification.” (This reinforces how the framework assesses uncertainty and clarifies what “optimization against observed data” entails.)

Line 113: In line with the above comment, the notion of “test observations” is not clearly defined.

See response above. We have also added the following details to the revised text:

“The Bayesian optimisation generates probability distributions for each simulated tree cover scenario, providing comprehensive uncertainty quantification^{27,28}. This combination of systematically isolating the influence of each control and the Bayesian technique’s production of probability distributions allows us to assess how confident we are in the model’s prediction”

Also, I found supplementary figure 2 very hard to assess. I understand now that on the x-axis are NME values, and y-axis is frequency (Basically these are histograms?)

Yes, the y-axis is a very accurate approximation of the posterior distribution. It was generated using 25,000 samples, as stated in the method, and plotted as a histogram. This is a standard way of plotting a distribution, but I have revised the caption to make it clear that the histogram and distribution are both accurate terms:

Supplementary Figure 2. NME step1-3 metrics scores. Top: the frameworks posterior (*histogram, generated from 25,000 samples*) and median (dashed line), mean (dot-dashed line), and randomly resampled (black) null models. Dumbell on the x-axis shows the x-axis range on subsequent rows, which display scores for individual rainfall distribution metrics coloured by rainfall and burnt area datasets.

but what step 1-3 refers to, and how exactly $\text{sim}(\beta)$ links to $P(X|\beta)$ remains unclear from the figure caption or the above description.

Step 1-3 describes the three NME comparisons, as outlined in Kelley et al. (2013, 2021):

- Step 1: Compares the area-weighted absolute difference between model ($\text{sim}(\beta)$) and observed values, normalizing by the mean variation in observations. This step captures the mean difference between observations and model simulations.

- Step 2: Removes the mean bias from both model and observations before calculating the NME, allowing the comparison to focus on shape differences between the datasets rather than any overall offset.
- Step 3: Further removes absolute variance from both model and observations. This final step ensures that differences are purely in spatial patterns and not due to differing variability between the two datasets.

By using these three steps, we assess not only the overall fit between model and data (Step 1), but also how well the model simulates patterns independent of mean bias (Step 2) and variance (Step 3). We have added a full description of NME steps 1-3 in the “Benchmarking” section of the SI.:

The NME calculation occurs in three steps:

NME Step 1: Initial comparison between model and observations.

The first step computes the NME based on the absolute area-weighted difference between observed and modelled tree cover values:

$$NME_1(\{obs_i\}, \{sim(\beta)_i\}) = \frac{\sum_i A_i \times |sim(\beta)_i - obs_i|}{\sum_i A_i \times |\overline{obs} - obs_i|}$$

NME Step 2: Removal of mean bias.

The second step eliminates the mean bias from both observations and model simulations by subtracting the mean from the data:

$$NME_2(\{obs_i\}, \{sim(\beta)_i\}) = NME_1(\{obs_i - \overline{obs}\}, \{mod_i - \overline{sim(\beta)_i}\})$$

Where $\overline{sim(\beta)_i}$ is the area-weighted mean of the simulated values. This step adjusts for systematic biases, allowing a comparison based solely on deviations from the mean.

NME Step 3: Normalization by variance.

The final step normalizes both observations and model values by their absolute variance:

$$NME_3(\{obs_i\}, \{sim(\beta)_i\}) = NME_1\left(\frac{\{obs_i - \overline{obs}\}}{V(\{obs_i\})}, \frac{\{mod_i - \overline{sim(\beta)_i}\}}{V(\{sim(\beta)_i\})}\right)$$

$$\text{Where } V(\{x_i\}) = \frac{\sum_i A_i \times |x_i - \overline{x}|}{\sum_i A_i}$$

This step standardizes the data, enabling a more precise comparison of variability between the modelled and observed tree cover.

We have also added “When performing comparisons in Supplementary Fig. 2, the parameters β , used in $sim(\beta)$, are drawn from the model's posterior parameter distribution” to clarify where the β for sim were sampled from.

Line 165: The Caption from supplementary Table 1 is confusing. The legend/caption of this table confuses me. If I read this, it for example means that the probability that the impact of burnt area (in the row) exceeds the impact of MAP (in the column) = 1. This means Fire has a BIGGER impact, not a smaller impact as suggested in the text. Or the explanation for this table is not correct (for example, these values represent P-values, and hence < 0.05 is the norm?)

Apologies. “row” and “column” were the wrong way around in the caption. We have now fixed this and the caption reads: “Supplementary Table 1: Probability of tropics-wide impact of variable in the column exceeding the impact of variable in the row.”

Page 22 Figure 3, similar issue as with previous case. Perhaps provide an example to make the figure understandable. So for the first map, is it correct that the impact of heat stress is almost 100% more likely than the impact of burnt area? Or is it the reverse?

The caption is correct, and note we have now expanded it in response to reviewer 1. There is also a small legend for each map indicating which stress the colour refers to, and we have added "The stress or pressure's first two letters or initials are listed next to the relevant colour for each map. For example, the top left blue areas show where Hs (Heat stress) reduces tree cover more than Ba (Burnt area)" to the caption.

Line 449 Figure 4, Make the titles in the figure better. Now the top one on the left reads just "tree cover" (probably evergreen?) and the one on the right reads "Deciduous cover", but without "fire impact on.."in it. Try to make this consistent with the description below for ease of interpretation.

The left-hand colours show how much fire reduces tree cover. The right shows the deciduous proportion of tree cover. To be more specific, we now use these in the figure title. We do not show the evergreen cover on the figure, but that is just the reciprocal of deciduous cover, so will be the green areas on the right.

Line 529: This equation does not represent a logistic curve at all, and also does not match the way it is represented in the references article (ref 27).

The logistic curve is outlined in ref 27, though we now include the equation in the methods section:

Each control was expressed as a linear combination of its respective factors. Fractional tree cover (TC) was calculated as a product of limitations imposed by control ($f(k_c \times (X_c - X_{0,c}))$) where c is a control ($C = \{MAP_*, MAT_*, SW, S, LU\}$), with each control's limitation represented by a logistic curve, f :

$$TC = TC_{max} \times \prod_c^c f(k_c \times (X_c - X_{0,c}))$$
$$f(x) = 1/(1 + e^{-x}) \quad (1)$$

Line 535: when you perform a log transformation, approaching 0 goes to negative infinity, it is not clear to me how that corresponds to making equation 1 go to 0 unless the assumption stated implies that there are no values of 0, but the lowest value for MAP is 1. If you mean including an infinitely negative exponent in the logistic equation (which is not what we currently see in Eq1), then indeed you would go to 0. But then this needs to be corrected.

We use the transformation outlines in Kelley et al. (2019) and adapt the text to read "MAP (i.e. $MAP_* = \log(MAP + 1/n_{cells})$ where n_{cells} is the number of grid points)". This ensures we do not get negative infinities. However, this is mainly for computational reasons. Analytically the limit of $\log(x)$ as $x \rightarrow 0$ is perfectly fine in equation 1, as the means the logistic function return a value of 0 when $k > 0$ or 1 when $k < 0$.

Line 543: It reads "...for $f=SW, LU, p[i]=1$ ". I Assume, that you mean $p[i] \neq 1$ (or perhaps $p[i] > 1$) for S?

No, what we have written is correct. The power is applied to stress only, as stated in the text. Using $p_i = 1$ means that we are effectively not raising the variable to a power for Land Use and Short Wave controls.

Line 553: The caption of supplementary figure 6 is not grammatically correct

We have re-worded to say "Supplementary Figure 6. The five annual burnt area products used in this study from the Fire Model Intercomparison Project16. See Supplementary Table 2 for dataset reference."

Line 555: there is no supplementary figure 12. I assume you mean figure 8

Corrected to supplementary figure 8

Line 577: What kind of resampling was done? Average? Maximum? Nearest Neighbour? Average may seem a logical choice, but I could also imagine that using the maximum is quite a good way, as this may have a much bigger impact on tree cover than the average. So this also needs a justification

We chose to use bilinear interpolation, now added to the text, for resampling because it allows for a smooth and continuous transition between grid points by averaging the four nearest neighbouring grid cells. This method aligns with the main goal of our study, which is to test assumptions and provide observational constraints on the effects of disturbances within dynamic global vegetation models (DGVMs). Bilinear interpolation maintains the concept of grid-cell averaging, ensuring that the resampled values accurately represent the overall landscape without introducing artificial extremes that may not align with the underlying assumptions of the models. While selecting the maximum value could emphasize more extreme impacts on tree cover, it would not be consistent with the general approach of DGVMs, which aim to simulate ecosystem processes at broader spatial scales using average conditions

In contrast, for certain climate variables that are more sensitive to extremes—such as wind speed and maximum temperature—we used the maximum value in the temporal dimension (i.e., maximum monthly) to capture these critical drivers of ecosystem disturbance.

Line 626-626: there seems to go something wrong with the equation formatting (perhaps mixup between word-equations and lateX-type equation formatting), but this needs to be cleaned up. This accounts for multiple equations. For example equation 9 has too few right-sided brackets etc.

The formatting issues highlighted are now fixed (we hope!)